


# Effects of LGM sea surface temperature and sea ice extent on the isotope-temperature slope at polar ice core sites

Alexandre CAUQUOIN[1], Ayako ABE-OUCHI[2], Takashi OBASE[2], Wing-Le CHAN[3], André PAUL[4] and Martin WERNER[5]

[1]Institute of Industrial Science (IIS), The University of Tokyo, Kashiwa, Japan
[2]Atmosphere and Ocean Research Institute (AORI), The University of Tokyo, Kashiwa, Japan
[3]Research Center for Environmental Modeling and Application, Japan Agency for Marine-Earth Science and Technology (JAMSTEC), Yokohama, Japan
[4]MARUM – Center for Marine Environmental Sciences and Department of Geosciences, University of Bremen, Bremen,
Germany
[5]Alfred Wegener Institute (AWI), Helmholtz Centre for Polar and Marine Sciences, Bremerhaven, Germany

*Correspondence to*: Alexandre Cauquoin (cauquoin@iis.u-tokyo.ac.jp)

**Abstract.** Stable water isotopes in polar ice cores are widely used to reconstruct past temperature variations over several orbital climatic cycles. One way to calibrate the isotope-temperature relationship is to apply the present-day spatial relationship as a
surrogate for the temporal one. However, this method leads to large uncertainties because several factors like the sea surface conditions or the origin and the transport of water vapor influence the isotope-temperature temporal slope. In this study, we investigate how the sea surface temperature (SST), the sea ice extent and the strength of the Atlantic Meridional Overturning Circulation (AMOC) affect these temporal slopes in Greenland and Antarctica for Last Glacial Maximum (LGM, ~21 000 years ago) to preindustrial climate change. For that, we use the isotope-enabled atmosphere climate model ECHAM6-wiso,
forced with a set of sea surface boundary condition datasets based on reconstructions (e.g., GLOMAP) or MIROC 4m simulation outputs. We found that the isotope-temperature temporal slopes in East Antarctic coastal areas are mainly controlled by the sea ice extent, while the sea surface temperature cooling affects more the temporal slope values inland. Mixed effects on isotope-temperature temporal slopes are simulated in West Antarctica with sea surface boundary conditions changes, because the transport of water vapor from the Southern Ocean to this area can dampen the influence of temperature on the
changes of the isotopic composition of precipitation and snow. In the Greenland area, the isotope-temperature temporal slopes are influenced by the sea surface temperatures very near the coasts of the continent. The greater the LGM cooling off the coast of southeast Greenland, the larger the temporal slopes. The presence or absence of sea ice very near the coast has a large influence in Baffin Bay and the Greenland Sea and influences the slopes at some inland ice cores stations. We emphasize that the extent far south of the sea ice is not so important. On the other hand, the seasonal variations of sea ice distribution, especially
its retreat in summer, influence the water vapor transport in this region and the modeled isotope-temperature temporal slopes in the eastern part of Greenland. A stronger LGM AMOC decreases LGM to preindustrial isotopic anomalies in precipitation in Greenland, degrading the isotopic model-data agreement. The AMOC strength does not modify the temporal slopes over



inner Greenland, and only a little on the coasts along the Greenland Sea where the changes in surface temperature and sea ice distribution due to the AMOC strength mainly occur.

## 1 Introduction

Stable isotopologues of water ($H_2^{16}O$, $H_2^{18}O$ and $HD^{16}O$, called hereafter stable water isotopes) are integrated tracers of climate processes occurring in diverse parts of the hydrological cycle (Craig and Gordon, 1965; Dansgaard, 1964). Because of their differences in mass and symmetries, an isotopic fractionation happens at each phase change of water. This process is reflected by a change in the water isotope ratio values, expressed hereafter in the usual δ notation (as $\delta^{18}O$ and $\delta^2H$ with respect to the Vienna Standard Mean Ocean Water V-SMOW if not stated otherwise). As a result, water isotopes have been widely used to describe past variations of the Earth's climate. For example, their measurements in polar ice cores made it possible to reconstruct the temperature variations over several glacial-interglacial cycles (Jouzel et al., 2007; Jouzel, 2013, and references therein; NEEM Community Members, 2013).

For such a reconstruction, the present-day isotope-temperature spatial slope can be taken as a surrogate for the temporal gradient at a given site. For example, a spatial slope of 0.80 ‰ °C$^{-1}$ for $\delta^{18}O$ in Antarctica was calculated based on a compilation of measured surface temperatures and $\delta^{18}O$ of snow at various locations in the continent (Masson-Delmotte et al., 2008). However, this method often leads to a large error in the temperature reconstructions because the temporal isotope-temperature slope depends on many factors like the sea surface temperature (SST) (Risi et al., 2010), the sea ice extent (Noone and Simmonds, 2004), the ice sheet elevation (Werner et al., 2018), the origin and the transport of water vapor (Casado et al., 2018). For example, it has been suggested that the relationship between temperature and the isotopic signature for warmer interglacial periods in East Antarctica can vary among ice core sites, with an error in the temperature reconstruction that can reach up to 100 % (Sime et al., 2009; Cauquoin et al., 2015). In Greenland, the use of the spatial relationship between the $\delta^{18}O$ in Greenland ice core records and surface temperature to evaluate the local temperature variations during the last deglaciation leads to a large uncertainty of a factor of 2 (Jouzel, 1999; Buizert et al., 2014). Recently, Buizert et al. (2021) proposed a reconstruction of surface cooling in Antarctica during the Last Glacial Maximum (LGM, ~21 000 years ago) using borehole thermometry and firn properties of different ice cores. Based on these results, they proposed new estimates of temporal $\delta^{18}O$-temperature slopes at these ice core stations, varying from 0.8 to 1.45 ‰ °C$^{-1}$.

The LGM is a period with full glacial conditions and represents the beginning of the last deglaciation. It is one of the key climate periods chosen by the Paleoclimate Modeling Intercomparison Project (PMIP, Kageyama et al., 2018, 2021) because it allows to evaluate how well state-of-the-art models are able to simulate climate changes as large as those expected in the future. In addition to being very different from the preindustrial climate (PI), the LGM period also offers a wealth of isotope proxy data, including stable water isotopes in polar ice cores for an in-depth comparison with outputs from isotope-enabled models (Lee et al., 2008; Risi et al., 2010; Werner et al., 2016, 2018).



One way to capture the physical processes influencing the temporal isotope-temperature slope in polar regions is the use of
Atmospheric General Circulation Models (AGCMs) equipped with prognostic stable water isotopes. Such models can simulate
different climate conditions, like LGM and PI periods. Moreover, the use of isotope-enabled AGCMs in combination with
isotopic observations allows us to investigate the physical processes controlling the variations of isotopic delta values at a
given site. This method makes it possible to estimate the temporal isotope-temperature slope for LGM to preindustrial climate
change (Lee et al., 2008; Risi et al., 2010; Werner et al., 2018). Even if such models simulate various temporal isotope-
temperature slopes, implying that processes like water vapor transport, post-depositional effects, or polar atmospheric
boundary layer are poorly or not represented (Krinner et al., 1997; Werner et al., 2000; Casado et al., 2018), these models are
very useful for evaluating the sensitivity of the temporal slopes to parameters like the change of elevation (Werner et al., 2018).
Ocean surface conditions are one of the factors that influences LGM-PI isotope changes (Risi et al., 2010; Noone and
Simmonds, 2004). Two reconstructions of SST and one of sea ice extent during the LGM period have been released recently.
Paul et al. (2021) reconstructed both the SST and the sea ice extent fields, based on faunal and floral assemblage data of the
Multiproxy Approach for the Reconstruction of the Glacial Ocean Surface (MARGO) project and several recent estimates of
the LGM sea ice extent. The Data-Interpolation Variational Analysis (DIVA) software was used to optimally interpolate sparse
SST reconstruction data. The resulting reconstruction was called GLOMAP (Glacial Ocean Map). Tierney et al. (2020)
reconstructed the LGM SST field with a different method, by combining a large collection of geochemical proxies for sea
surface temperature with simulations outputs from the isotope-enabled model iCESM1.2 (Brady et al., 2019) using an offline
data assimilation technique to produce a field reconstruction of LGM temperatures. Tierney et al. (2020) LGM cooling is
globally larger than in GLOMAP (3.6°C and 1.7°C, respectively), with possible impacts on LGM to PI isotope changes and
their temporal relationship with near surface air temperature. In addition, other SST and sea ice fields, with different
characteristics compared to the reconstructions of LGM sea surface conditions described above, can be extracted from
atmosphere-ocean coupled model simulations like MIROC 4m (Obase and Abe-Ouchi, 2019).

In the present study, we investigate the impacts of SST and sea ice boundary conditions on the isotope-temperature temporal
slope at polar ice core sites for LGM-to-PI changes. For that, we performed multiple simulations with the isotope-enabled
AGCM ECHAM6-wiso driven by different LGM SST and sea ice boundary conditions. We evaluate the modeled LGM-PI
$\delta^{18}O$ anomalies with available observations and we investigate how the SST and the sea ice extent patterns influence the model-
data agreement on a global scale and at polar ice core stations. The influence of ocean circulation, particularly the strength of
the Atlantic Meridional Overturning Circulation (AMOC), on sea surface conditions and by extension on our modeled $\delta^{18}O$ of
meteoric water is also investigated. Finally, the impacts of the sea surface boundary conditions on the $\delta^{18}O$-temperature slopes
for LGM-to-preindustrial climate change are evaluated and discussed for Greenland and Antarctic ice core stations.



## 2 Methodology

### 2.1 ECHAM6-wiso

ECHAM6 (Stevens et al., 2013) is the sixth generation of the atmospheric general circulation model ECHAM, developed at the Max Planck Institute for Meteorology. It consists of a dry spectral-transform dynamical core, a transport model for scalar quantities other than temperature and surface pressure, a suite of physical parameterizations for the representation of diabatic processes, and boundary datasets for externalized parameters (trace gas and aerosol distributions, land surface properties, etc.). ECHAM6 forms the atmospheric component of the fully coupled Earth system model MPI-ESM (Giorgetta et al., 2013; Mauritsen et al., 2019). The implementation of the water isotopes in ECHAM6 as part of MPI-ESM has been described and evaluated in detail by Cauquoin et al. (2019b), and this model version has been labeled ECHAM6-wiso. At a later stage, Cauquoin and Werner (2021) updated the water isotope module of ECHAM6-wiso in several aspects. The supersaturation has been slightly re-tuned, the kinetic fractionation factors for the evaporation over the ocean are now assumed as independent of wind speed, and the isotopic content of snow on sea ice is taken into account for sublimation processes in sea ice covered regions. The latter leads to a stronger depletion of surface water vapor over such sea ice covered areas (while the surface temperature remains the same). As a consequence, this change is expected to contribute to a steeper temporal isotope-temperature slope over sea ice covered areas.

### 2.2 Sea surface temperature and sea ice extent boundary conditions for LGM conditions

#### 2.2.1 SST

Tierney et al. (2020) SST reconstruction has a larger and more homogeneous cooling than GLOMAP, except for the high southern latitudes at which the Pacific sector cools more than the Atlantic sector (Figure 1). On the other hand, the LGM cooling in the Northern North Atlantic Ocean is stronger in GLOMAP than in Tierney et al. reconstruction (-5.4°C and -4.8°C, respectively, see Table 4 in Paul et al., 2021). These differences between the two SST reconstructions are due to the use of different proxy datasets for the reconstructions (geochemical proxies only for Tierney et al., MARGO dataset for GLOMAP) and to the methods applied to produce SST gridded maps from scattered observations (see Section 1). For their offline data assimilation technique, Tierney et al. (2020) used results from the coupled climate model iCESM1.2, which shows one of the largest cooling among the PMIP4 models (Figure 1b of Kageyama et al., 2021). In addition to these two reconstructions, we used SST and sea ice extent outputs from a MIROC 4m LGM simulation (Obase and Abe-Ouchi, 2019) with oscillating AMOC strength. The global LGM cooling is between -2.3 and -2.7°C according to the considered simulations (Figure 1) i.e., higher than GLOMAP and lower than the Tierney et al. reconstruction. The main specificity of MIROC 4m LGM SST is a very strong cooling in the North Atlantic (more than 10°C, Figure 1) and more uniform temperature anomalies between -2 and -4°C in the other areas, including off the coast of Greenland. We extracted the MIROC 4m SST outputs, averaged over a 100-year period, at two different times of the LGM simulation depending on the AMOC strength: during a weak AMOC phase (average AMOC index was equal to 8.44 Sv) and a strong AMOC phase (19.95 Sv). A weaker AMOC during LGM implies larger cooling in





the North Atlantic (Figure 1) and more extended sea ice (Figure 2), while it does less cooling in the Southern Ocean. The strong AMOC phase period in MIROC 4m simulation was selected in the middle of the AMOC peak. Therefore, the values of MIROC 4m average near surface air temperature in Antarctica are very similar regardless the selected AMOC phase. For example, MIROC 4m simulates LGM temperature of -41.87 and -41.75°C in WDC station for strong and weak AMOC phase,

respectively. A similar pattern is found for the eastern part of the continent (-56.80 and -56.50°C in Dome Fuji for strong and weak AMOC phase, respectively).

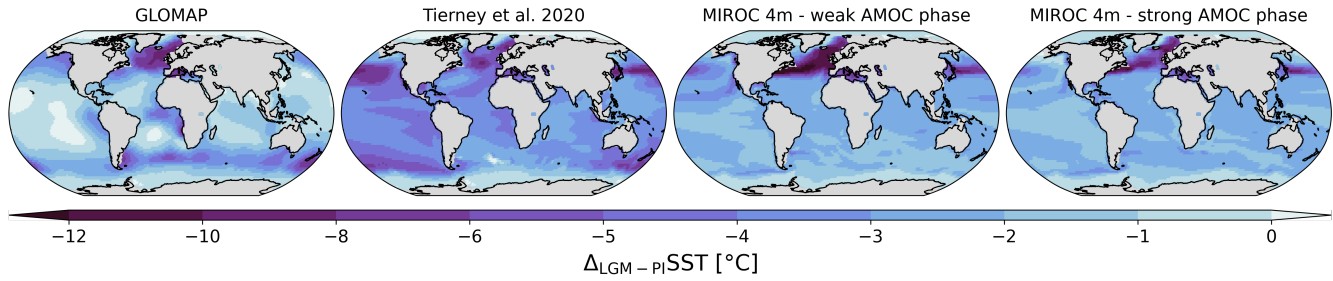

**Figure 1: LGM-PI sea surface temperature changes used as boundary conditions for ECHAM6-wiso simulations. From left to right: GLOMAP (Paul et al., 2021), Tierney et al. (2020), MIROC 4m with weak LGM AMOC phase and MIROC 4m with strong LGM**
**AMOC phase.**

### 2.2.2 Sea ice extent

Maps of the averaged sea ice area fraction used as boundary forcings for ECHAM6-wiso are shown in Figure 2. The PI AMIP and LGM GLOMAP sea ice cover is higher around Antarctica compared to MIROC 4m ones, with a further extent in the Southern Ocean especially in the Atlantic sector. On the other hand, sea ice is more extensive in the Northern Hemisphere for

MIROC 4m in the weak AMOC phase. For the stronger AMOC case, a decline of the sea ice in the Northern Hemisphere is seen, accompanied by weaker cooling (see section 2.2.1). In its parameterization, MIROC 4m uses a threshold of 95% for the sea ice fraction to allow sub-grid "sea-ice leads". This threshold is not rigid, but it is difficult to exceed sea ice concentrations of 95% unless there is significant convergence of sea-ice. Consequently, while the sea ice is, on average, more extensive in the north in MIROC 4m for the weak AMOC phase compared to GLOMAP reconstruction, the sea ice area fraction in grid

cells near coastal areas like Greenland is lower in MIROC 4m than in GLOMAP (95-98% against 100%, respectively).



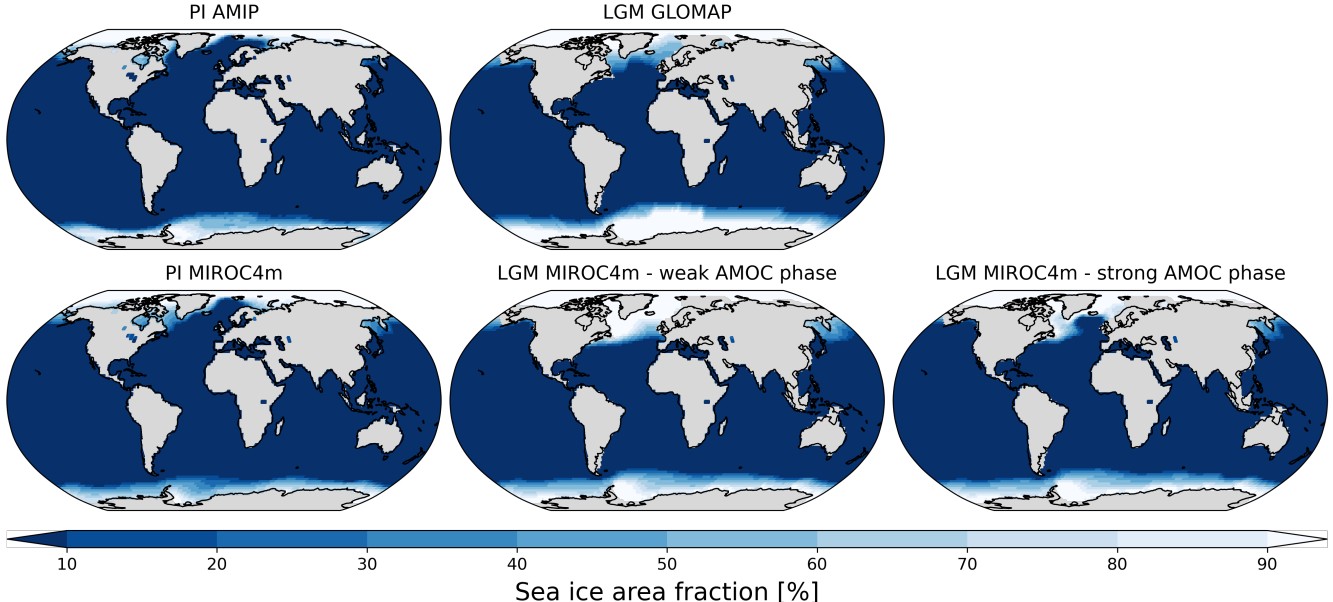

**Figure 2: LGM and PI sea ice area fractions used as boundary conditions for ECHAM6-wiso simulations.**

## 2.3 Model setup and experiments

We performed an ensemble of LGM simulations with ECHAM6-wiso, forced with different combinations of SST and sea ice
boundary forcings presented in section 2.2. The LGM SST boundary fields are expressed relative to the AMIP mean SST
(averaged over the period 1870 to 1899) used for the preindustrial simulations. The GLOMAP reconstruction has the advantage
of providing a monthly climatology of LGM SST and sea ice extent, while only annual mean SST is available from the
reconstruction by Tierney et al. (2020), without sea ice map distribution. So, Tierney et al. LGM SST for ECHAM6-wiso was
produced by taking the annual mean SST anomaly from Tierney et al. (2020) and adopting the monthly climatology
temperature variability from GLOMAP. We used the sea ice extent data from GLOMAP in this case, too. In order to investigate
the impact of sea ice extent on our isotope results and the related isotope-temperature slopes for LGM-to-PI climate change,
we used LGM SST outputs from MIROC 4m simulations combined with sea ice extent data from the same MIROC 4m
simulations or from GLOMAP dataset. Similarly for PI conditions, we performed several PI simulations with different sea ice
boundary conditions depending on the setup of LGM experiments, using climatological monthly mean sea ice area fractions
from AMIP or MIROC 4m coupled simulations. The prescribed LGM ice sheet is GLAC-1D (Tarasov and Peltier, 2002;
Tarasov et al., 2012, 2014; Abe-Ouchi et al., 2013; Briggs et al., 2014) for all LGM simulations. As with SST and sea ice
distribution, mean $\delta^{18}O$ of surface seawater needs to be prescribed. For the PI simulations, we used the $\delta^{18}O$ reconstruction
from the global gridded data set of LeGrande and Schmidt (2006). As no equivalent data set of the $\delta^{2}H$ composition of seawater
exists, the deuterium isotopic composition of the seawater in any grid cell has been set equal to the related $\delta^{18}O$ composition,
multiplied by a factor of 8, in accordance with the observed relation for meteoric water on a global scale (Craig, 1961). As in





Werner et al. (2018), a prescribed glacial seawater enrichment of +1 ‰ and +8‰ is assumed for $\delta^{18}O$ and $\delta^2H$ in the LGM simulations, respectively. Finally, the greenhouse gas and orbital conditions were prescribed according to PMIP4 protocol. The PI and LGM simulations were run for 60 and 120 model years, respectively, and we used the last 30 model years for our analyses. The simulations' characteristics are summarized in Table 1. Two additional sensitivity simulations have been

performed to evaluate the impacts of lower MIROC 4m sea ice area fraction in coastal grid cells (section 2.2.2) and the consideration of the isotopic composition of snow on sea ice in ECHAM6-wiso (section 2.1) on the modeled $\delta^{18}O_P$-temperature temporal slopes between LGM and PI (see text in Supplementary Material). Also, a LGM simulation using the PMIP3 ice sheet reconstruction instead of GLAC-1D (see Figures 3b and 3d of Werner et al. (2018), respectively) has been performed to evaluate the impact of ice sheet topography on the isotopically enriched bias in Antarctica (see text in Supplementary Material).

**Table 1: Characteristics of the ECHAM6-wiso simulations in the present study.**

| LGM simulation name | SST | Sea ice | PI control simulation characteristics | Comments |
|---|---|---|---|---|
| LGM_GLOMAP | GLOMAP | GLOMAP | Mean PI SST and sea ice from AMIP | Lower SST cooling |
| LGM_tierney2020 | Tierney et al., 2020 | GLOMAP | Mean PI SST and sea ice from AMIP | Higher SST cooling |
| LGM_miroc4m_sst_glomap_sic | MIROC 4m | GLOMAP | Mean PI SST and sea ice from AMIP | AMOC oscillation: weak phase |
| LGM_miroc4m_sst_and_sic | MIROC 4m | MIROC 4m | Mean PI SST from AMIP and PI sea ice from MIROC 4m | AMOC oscillation: weak phase |
| LGM_miroc4m_strong_AMOC_sst_glomap_sic | MIROC 4m | GLOMAP | Mean PI SST and sea ice from AMIP | AMOC oscillation: strong phase |
| LGM_miroc4m_strong_AMOC_sst_and_sic | MIROC 4m | MIROC 4m | Mean PI SST from AMIP and PI sea ice from MIROC 4m | AMOC oscillation: strong phase |

## 2.4 Observational data

To evaluate the modeled $\delta^{18}O$ of precipitation and snow values at ice core stations, we use here a selection of 6 Greenland and 10 Antarctic ice cores for the preindustrial and LGM climates (Figure 3). The observed $\delta^{18}O$ values were defined as averages over the last 200 years for the preindustrial period, and in the $21 \pm 1$ ka period for the LGM. We also use LGM-PI $\delta^{18}O$

anomalies from 5 (sub-)tropical ice cores that are reported in Table 2 of Risi et al. (2010). The ice core data used in this study are summarized in Table 2. In order to mitigate the seasonal bias when comparing observed $\delta^{18}O$ from snow in ice cores with modeled $\delta^{18}O$ of precipitation or deposited snow, the modeled $\delta$ values are calculated as a precipitation (or snow)-weighted mean with respect to the V-SMOW scale. For the evaluation of modeled $\delta^{18}O$ of precipitation at a global spatial scale, we extracted 14 entities from the SISALv2 speleothem dataset (Comas-Bru et al., 2020) where both PI and LGM $\delta^{18}O$ values of

calcite or aragonite are available. As recommended by Comas-Bru et al. (2019), we defined here averaged PI and LGM values as the means of the 1850-1990 CE and $21 \pm 1$ ka periods, respectively. To compare the $\delta^{18}O$ of speleothem data with our modeled $\delta^{18}O$ of precipitation ($\delta^{18}O_P$), the measured $\delta^{18}O$ of calcite or aragonite are converted into $\delta^{18}O$ of drip-water using equations 1 or 2 of Comas-Bru et al. (2019), respectively, after conversion from V-PDB to VSMOW scale (equation 3 of





Comas-Bru et al. (2019)). The annual mean surface air temperature from ECHAM6-wiso is used for the conversion. A seasonal
bias can appear in the isotopic composition of drip water archived in speleothem records due to the re-evaporation of the
precipitated water (Wackerbarth et al., 2010). An additional fractionation between the drip water and the formed
calcite/aragonite can also be observed for many speleothems (Dreybrodt and Scholz, 2011).

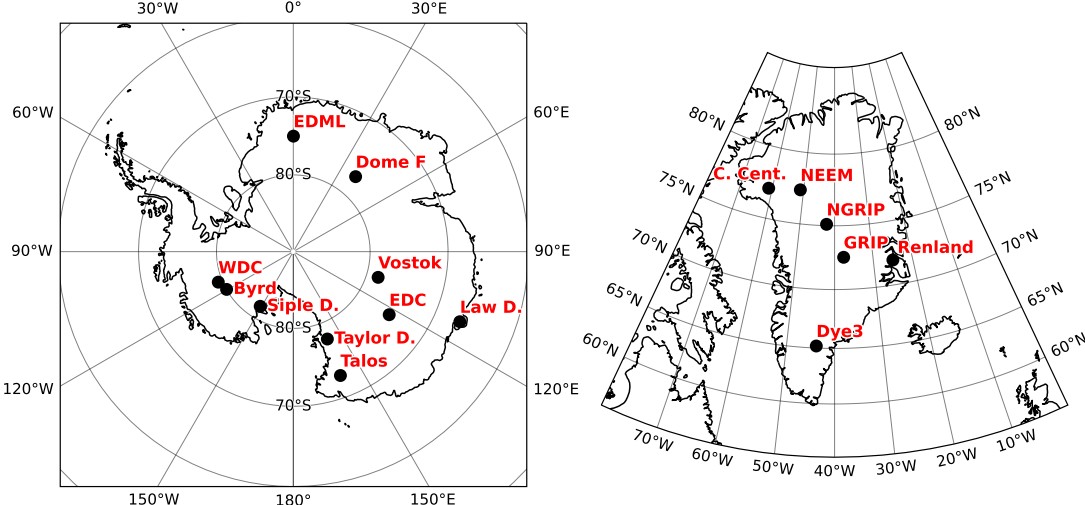

**Figure 3: Location of polar ice core sites in Antarctica (left) and Greenland (right).**









**Table 2: Selected ice cores records and their geographical coordinates, reported PI values of δ¹⁸O and changes in δ¹⁸O between LGM and PI.**

| Site | Longitude | Latitude | $\delta^{18}O_{PI}$ (‰) | $\Delta_{LGM-PI}\delta^{18}O$ (‰) |
|---|---|---|---|---|
| Vostok[a,b] | 106.87 | -78.47 | -56.8 | -4.8 |
| Dome F[c] | 39.70 | -77.32 | -54.6 | -4.9 |
| EDC[d,e] | 123.35 | -75.10 | -50.4 | -5.6 |
| EDML[b,d] | 0.07 | -75.00 | -44.8 | -6.3 |
| Law Dome[b] | 112.83 | -66.73 | -22.4 | -5.5 |
| Taylor Dome[f] | 158.72 | -77.8 | -40.5 | -3.5 |
| Talos[g] | 159.18 | -72.82 | -36.1 | -5.4 |
| Byrd[h] | -119.52 | -80.02 | -32.9 | -7.3 |
| Siple Dome[b] | -148.82 | -81.67 | -25.6 | -7.8 |
| WDC[b] | -112.14 | -79.46 | -34 | -7.3 |
| GRIP[a,j] | -37.63 | 72.58 | -35.3 | -5.4 |
| NGRIP[a,k] | -42.32 | 75.10 | -35.2 | -7.4 |
| NEEM[l,m] | -51.06 | 77.45 | -33 | -10 |
| Camp Century[i] | -61.13 | 77.17 | -29.3 | -12.9 |
| Dye3[j] | -43.81 | 65.18 | -27.7 | -7.3 |
| Renland[i] | -25.00 | 72.00 | -27.4 | -3.8 |
| Huascaran[a] | -77.61 | -9.11 | - | -6.3 |
| Sajama[a] | -68.97 | -18.1 | - | -5.4 |
| Illimani[a] | -67.77 | -16.62 | - | -6 |
| Guliya[a] | 81.48 | 35.28 | - | -5.4 |
| Dunde[a] | 96 | 38 | - | -2 |

References: [a] reported in Risi et al. (2010), [b] WAIS Divide project members (2013), [c] Kawamura et al. (2007), [d] Stenni et al. (2010), [e] Landais et al. (2015), [f] Steig et al. (2000), [g] Stenni et al. (2011), [h] Blunier and Brook (2001), [i] Vinther et al. (2009), [j] Vinther et al. (2006), [k] North Greenland Ice Core project members (2004), [l] Guillevic et al. (2013), [m] Schüpbach et al. (2018).

## 3 Results of the LGM-PI ECHAM6-wiso simulations

### 3.1 Evaluation of ECHAM6-wiso under LGM conditions

We evaluate here the global distribution of $\delta^{18}O_p$ changes between LGM and PI ($\Delta_{LGM-PI}\delta^{18}O_p$) from our different ECHAM6-wiso simulations. Figure 4 shows the comparison of modeled $\delta^{18}O_p$ anomalies with isotope measurements from ice cores and speleothems for the simulation LGM_miroc4m_sst_and_sic (i.e., SST and sea ice boundary conditions from MIROC 4m simulation at weak AMOC phase). Well-known patterns of global $\Delta_{LGM-PI}\delta^{18}O_p$ distribution are found in ECHAM6-wiso, like

the negative anomalies across Canada, Greenland and Northern Europe due to the presence of glaciers in these areas during





LGM period (Figure 4c). Generally, negative $\delta^{18}O_p$ anomalies are also simulated over Antarctica and the Southern Ocean due to lower temperatures in LGM compared to PI period (Figure 4a). In the mid-to-low latitudes, $\Delta_{LGM-PI}\delta^{18}O_p$ is mainly controlled by precipitation anomalies (Figure 4b). For example, lower modeled precipitation in the Amazonian area, over parts of South East Asia and in the western Pacific Ocean during the LGM leads to positive modeled $\delta^{18}O_p$ anomalies. Despite some biases

in modeled $\Delta_{LGM-PI}\delta^{18}O_p$, like in Southern Amazonia (Figures 4c and d) where negative anomalies are measured in ice cores (between -2 and -6‰, see green dots in Figure 4d) while positive anomalies are simulated (between 0 and 4‰), modeled $\delta^{18}O_p$ anomalies are in rather good agreement with observations from ice cores and speleothems (Figure 4d).

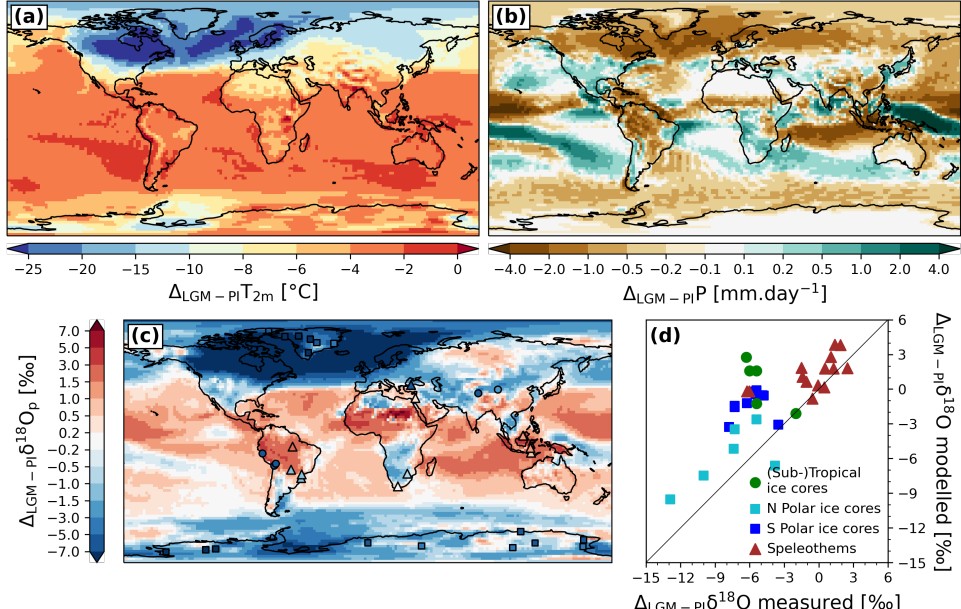

**Figure 4: Changes in modeled (a) 2m air temperature, (b) precipitation and (c) $\delta^{18}O_p$ between the LGM and PI climates from the**
**LGM_miroc4m_sst_and_sic simulation (background colors). In (c), the squares, dots and triangles represent $\delta^{18}O$ changes measured in polar ice cores, (sub-)tropical ice cores and speleothems, respectively. Measured $\delta^{18}O$ in calcite or aragonite from speleothems have been converted into $\delta^{18}O$ of drip-water before comparison with modeled $\delta^{18}O_p$ (see Section 2.4). (d) Scatter plot showing a comparison of observed $\delta^{18}O$ changes with modeled $\delta^{18}O_p$ anomalies at the nearest grid cell of the archives' locations. Northern and southern polar ice core locations are distinguished by cyan and blue colors, respectively.**

The isotope distribution is mainly controlled by changes in temperature and in the water cycle. Even though all the ECHAM6-wiso simulations show similar global distribution of 2m air temperature ($T_{2m}$) and precipitation responses to the various SST and sea ice boundary fields, we find some differences too (Figures S1 and S2 in Supplementary Material). As expected, the modeled global cooling using SST from GLOMAP is lower while it is stronger when using SST from Tierney et al. (2020) (cooling of -4 and -6.3°C, respectively). Average $T_{2m}$ anomalies in the middle range are obtained when using the MIROC 4m

SST fields (between -4.4 and -5.3°C depending on the MIROC 4m data used). The temperature over sea ice covered areas are largely impacted by the sea ice forcings used (i.e., GLOMAP or MIROC 4m). The modeled $T_{2m}$ anomalies over the Southern Ocean vary between -10 and -15°C with GLOMAP sea ice while the cooling is only between -4 and -10°C when ECHAM6-



wiso is forced by MIROC 4m sea ice. The opposite is true for the Arctic region. A strong cooling is simulated with the sea ice from MIROC 4m with a weak AMOC phase (a cooling of more than 20°C), more than with the sea ice from GLOMAP
(between -20 and -10°C). The different SST boundary conditions have a strong influence on the precipitation anomalies, especially at mid-to-low latitudes including the western Pacific area and the East Asian monsoon region (Figure S2). All these differences in $T_{2m}$ and precipitation responses have profound impacts on modeled $\delta^{18}O_p$ anomalies (Figure S3) and their agreements with observations (Figures 4c and d).

**Table 3: Values of $\Delta_{LGM-PI}\delta^{18}O$ model-data slope (1 is better), coefficient of determination $r^2$ and root mean square error (RMSE)**
**for our ECHAM6-wiso simulations using different SST and sea ice boundary fields. For each column, worst to best model-data agreements are shown with a yellow-to-green colormap.**

| LGM simulation name | Slope | $r^2$ | RMSE (‰) |
|---|---|---|---|
| LGM_GLOMAP | 0.699 | 0.540 | 3.943 |
| LGM_tierney2020 | 0.646 | 0.584 | 3.507 |
| LGM_miroc4m_sst_glomap_sic | 0.592 | 0.562 | 3.907 |
| LGM_miroc4m_sst_and_sic | 0.582 | 0.498 | 4.135 |
| LGM_miroc4m_strong_AMOC_sst_glomap_sic | 0.660 | 0.580 | 3.910 |
| LGM_miroc4m_strong_AMOC_sst_and_sic | 0.558 | 0.532 | 4.299 |

The statistics of $\Delta_{LGM-PI}\delta^{18}O_p$ model-data agreements are shown for our different ECHAM6-wiso simulations in Table 3. The best model-data agreement in terms of model-data slope (1 is perfect match) is found when using SST and sea ice from
GLOMAP (slope = 0.70) as boundary conditions for ECHAM6-wiso, but better coefficient of determination ($r^2$) and root mean square error (RMSE) are obtained with LGM SST from Tierney et al. (2020) ($r^2 = 0.58$ and RMSE = 3.5 ‰). We notice a worse model-data agreements in $\delta^{18}O_p$ changes when both SST and sea ice changes from MIROC 4m simulations are provided as sea surface boundary conditions (slopes lower than 0.582 and RMSE higher than 4.1 ‰). This is in agreement with Werner et al. (2018) who showed a worse model-data agreement when using SST and sea ice boundary conditions from a coupled
model instead of reconstructed ones. The substitution of MIROC 4m sea ice changes by GLOMAP ones improves the $\Delta_{LGM-PI}\delta^{18}O_p$ model-data agreement for all cases (i.e., weak or strong AMOC phase). For example, the model-data slope when using SST changes from MIROC 4m simulation during strong AMOC phase is similar to the one for the simulation with the Tierney et al. SST reconstruction (0.66 and 0.65, respectively).

**3.2 Impacts of SST boundary conditions on the $\Delta_{LGM-PI}\delta^{18}O$ model-data agreement at polar ice core stations**

The modeled values of $\Delta_{LGM-PI}\delta^{18}O$ of snow ($\delta^{18}O_{sn}$) at polar ice cores stations for different boundary conditions in LGM-PI SST changes are compared to isotopic observations in Figure 5a. Only simulations using sea ice from GLOMAP are selected here. Except for Renland station in the North and Taylor Dome in the South (that are both coastal sites), ECHAM6-wiso generally under-estimates $\delta^{18}O_{sn}$ changes. The main explanation for this general bias is the use of GLAC-1D ice sheet



reconstruction for our ECHAM6-wiso simulations. The substitution of GLAC-1D reconstruction by the PMIP3 one strongly
improves the model-data agreement of $\delta^{18}O$ in Antarctica (Figure S4), leading to a better model-data agreement at global scale
(slope = 0.87, $r^2$ = 0.62 and RMSE = 3.2 ‰) compared to the LGM_GLOMAP experiment. This agrees with the findings of
Werner et al. (2018) who showed that the isotopic model-data correlation for Antarctic ice core stations is weaker when using
GLAC-1D instead of PMIP3 ice sheet reconstruction (RMSE = 2.1 and 1.1 ‰ for 11 Antarctic stations, respectively). Except
for the Taylor Dome station, all modeled $\Delta_{LGM-PI}\delta^{18}O_{sn}$ at polar ice core stations are in better agreement with measurements
(blue bars in Figure 5a) when SST fields from GLOMAP or Tierney et al. are used (orange and green bars in Figure 5a,
respectively), confirming the results of Werner et al. (2018) about the worse model-data agreement when using sea surface
boundary conditions from a coupled model instead of reconstructed ones. The change from one MIROC 4m SST field to
another one (i.e., weak or strong AMOC phase) as input for ECHAM6-wiso does not modify the modeled $\Delta_{LGM-PI}\delta^{18}O_{sn}$ values
much (red and purple bars in Figure 5a).

As expected, the modeled cooling is globally lower when using SST from GLOMAP to drive ECHAM6-wiso (left maps in
subplots (c) to (e) of Figure 5). However, a strong cooling is obtained with GLOMAP SST in the Southern Ocean, which is
the evaporative source of isotopic signals measured in polar areas. As a consequence, temperature changes in Antarctica are
stronger when using SST from GLOMAP or Tierney et al., giving higher modeled $\delta^{18}O_P$ changes compared to modeled results
using SST fields from MIROC 4m (right maps of Figure 5), and better agreement with the observations. The stronger cooling
in the Atlantic sector of the Southern Ocean with GLOMAP SST compared to Tierney et al. one has the consequence of
enhancing the $\delta^{18}O_P$ depletion in the Atlantic-Indian Ocean sector of Antarctica (right map of Figure 5c) despite similar
temperatures between the two simulations (left map of Figure 5c). This area includes the ice core stations Dome Fuji and
EDML, and a better model-data agreement is found there when GLOMAP SST values are provided as boundary conditions
(Figure 5a). The opposite is true for other stations further to the east and west, like WDC and EDC. As in the Southern Ocean,
a higher cooling is simulated in the Northern North Atlantic Ocean to the south of Greenland if the SST from GLOMAP is
used. A stronger cooling is simulated in the southern and central part of inner Greenland, too (left maps of Figure 5). As a
consequence, higher $\delta^{18}O$ changes between LGM and PI are simulated in Greenland with the SST from GLOMAP (right maps
of Figure 5), except in Northern Greenland like at Camp Century station (Figure 5a). A better agreement is obtained with the
Greenland $\delta^{18}O$ observations under this configuration (orange bars in Figure 5a), except for Renland and Camp Century (worse
and similar model-data agreements among the simulation results, respectively).

**Figure 5: (a)** Comparison of modeled anomalies in $\delta^{18}O_{sn}$ between LGM and PI with $\delta^{18}O$ anomalies measured in polar ice cores **(blue bars). All modeled results are from simulations with the same sea ice boundary conditions from GLOMAP but with different SST forcings: GLOMAP (orange), Tierney et al. (2020) (green), MIROC 4m with weak LGM AMOC phase (red) and MIROC 4m with strong LGM AMOC phase (purple). (b) Modeled** $T_{2m}$ **and** $\delta^{18}O_p$ **changes between LGM and PI using GLOMAP SST (left and right maps, respectively). Maps in plots (c) to (e) show the impacts on** $T_{2m}$ **and** $\delta^{18}O_p$ **anomalies using the other SST boundary conditions. The values are expressed relative to the modeled results from (b).**



### 3.3 Impacts of sea ice changes boundary conditions on the $\Delta_{\text{LGM-PI}}\delta^{18}O$ model-data agreement at polar ice core stations

To analyze the effects of sea ice boundary conditions on the modeled $\delta^{18}O$ changes in polar regions between LGM and PI, we
compare the results from the simulations using the same SST (here from the MIROC 4m simulation with the weak AMOC phase) but different sea ice area fraction fields: GLOMAP and MIROC 4m (i.e., LGM_miroc4m_sst_glomap_sic and LGM_miroc4m_sst_and_sic simulations, respectively). For all Antarctic ice core stations, a stronger depletion in $\delta^{18}O_{sn}$ between LGM and PI is simulated with GLOMAP sea ice distribution (orange bars in Figure 6a). Except for Taylor Dome, a better agreement with isotopic observations is then found. The LGM sea ice from GLOMAP in the Southern Ocean is more
extensive than the one from MIROC 4m (Figure 2). It has a huge impact on modeled $T_{2m}$ anomalies over the Southern Ocean (between 2 and 10°C), and the simulated cooling is higher by 1 to 4°C in Western Antarctica and in coastal regions of the continent (left map of Figure 6c). As a consequence, higher LGM-PI anomalies in $\delta^{18}O$ of precipitation and of snow are simulated: more than 5 ‰ over the Southern Ocean and around 1-2 ‰ on the continent, especially in the western part (right map of Figure 6c). The situation is opposite to that of the Arctic Ocean and Greenland with the sea ice from MIROC 4m (weak
AMOC phase) being more extensive than the one from GLOMAP (Figure 2). Cooling is stronger by 5 to 10°C in the Arctic Ocean and from 0.5 to 5 °C in Greenland (left map of Figure 6c) when ECHAM6-wiso is forced by MIROC 4m sea ice boundary conditions, giving higher $\delta^{18}O_p$ anomalies of up to 2 ‰ (right map of Figure 6c). The cooling is slightly lower near the Greenland coast because the LGM-PI sea ice change is more important in GLOMAP compared to MIROC 4m. This is due to the lower sea ice area fraction in grid cells near coastal areas in MIROC 4m (95-98% against 100% in GLOMAP, see section
2.2.2). This lower sea ice change in MIROC 4m combined with the isotopic content of snow on sea ice taken into account for sublimation processes in sea ice covered regions leads to a reduction of the LGM-PI $\delta^{18}O_p$ changes in Baffin Bay (right map of Figure 6c). This aspect is investigated in detail in section 4.2. Finally, if the less extensive sea ice distribution from MIROC 4m under a strong AMOC phase is used to force ECHAM6-wiso, modeled $\delta^{18}O_{sn}$ changes at Greenland ice core locations become smaller than the ones with GLOMAP sea ice, weakening the model-data agreement for this region (Figure S5).





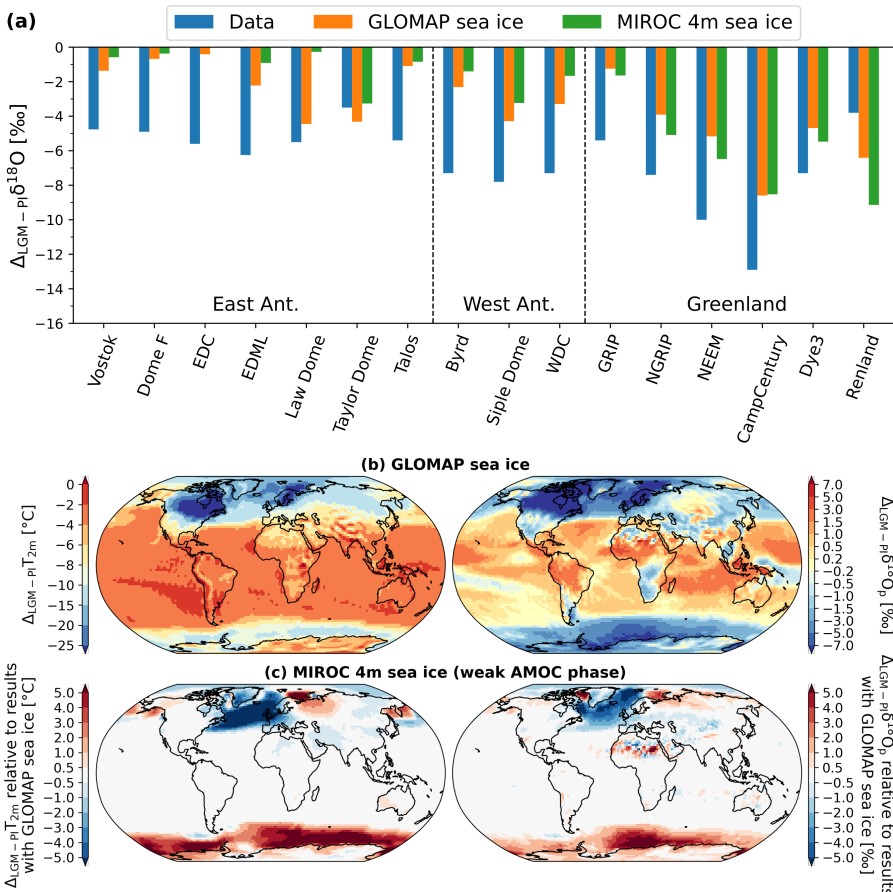

**Figure 6: (a) Comparison of modeled anomalies in $\delta^{18}O_{sn}$ between LGM and PI with $\delta^{18}O$ anomalies measured in polar ice cores (blue bars). Modeled results are from the simulations using the SST changes of MIROC 4m with weak LGM AMOC but different sea ice boundary conditions (GLOMAP and MIROC 4m with weak LGM AMOC phase in orange and green, respectively). (b) Modeled $T_{2m}$ and $\delta^{18}O_p$ changes between LGM and PI using GLOMAP sea ice (left and right maps, respectively). Maps in (c) show the impacts on $T_{2m}$ and $\delta^{18}O_p$ anomalies using sea ice from MIROC 4m instead. Values are expressed relative to the modeled results from (b).**

### 3.4 Impacts of LGM AMOC strength on the $\Delta_{LGM-PI}\delta^{18}O$ model-data agreement at polar ice core stations

Here, we investigate the impacts of AMOC strength on the modeled $\Delta_{LGM-PI}\delta^{18}O$ in polar regions. For that, sea surface outputs (i.e., both SST and sea ice spatial distribution) from the MIROC 4m simulation with different LGM AMOC strengths are used as boundary conditions for ECHAM6-wiso. We focus first on the North Pole region because the AMOC strength mainly influences the climate of the Northern Hemisphere, as shown in SST and sea ice distributions used in this study (Figures 1 and 2). A weaker AMOC during LGM involves less heat transported in the north and thus lower LGM temperatures (i.e., larger cooling relative to PI), as shown in the left map of Figure 7c. A difference in $T_{2m}$ of up to 10°C in the North Atlantic and Arctic Oceans is seen in the LGM_miroc4m_strong_AMOC_sst_glomap_sic and LGM_miroc4m_strong_AMOC_sst_and_sic simulations. Cooling in Greenland is reduced by 2-4 °C when the AMOC is increased. LGM to PI changes in $\delta^{18}O$ in Greenland





is mainly controlled by this change in mean temperature with an increase in LGM $\delta^{18}O_{sn}$ of between 1.2 and 2.5 ‰ at Greenland ice core stations for a stronger LGM AMOC (orange and green bar in Figure 7a). As ECHAM6-wiso generally underestimates the LGM-PI $\delta^{18}O$ changes at the poles, a weaker AMOC generally improves the model-data agreement (blue and orange bars in Figure 7a). In the Southern Ocean and Antarctic regions, only small $T_{2m}$ changes are simulated by ECHAM6-wiso due to a

345 change in AMOC strength during LGM (left map of Figure 7c). As a consequence, modeled $\Delta_{LGM-PI}\delta^{18}O_{sn}$ values are very similar between the 2 simulations (orange and green bars in Figure 7a). These small differences are due to the selection of the strong AMOC phase period in the middle of the peak in MIROC 4m simulation (see section 2.2.1). The impact of the period selection for the strong AMOC phase (e.g., the start or the end of the interstadial) on surface temperature and $\delta^{18}O$ in Antarctica will be investigated more in detail in a future study. Finally, the changes of SST values alone due to AMOC strength variations

change by only less than 1 ‰ of the modeled $\Delta_{LGM-PI}\delta^{18}O_{sn}$ (red and purple bars in Figure 5a). This shows that the LGM to PI changes in sea ice distribution, related to the AMOC strength variations, have a large impact on modeled $T_{2m}$ anomalies and consequently on the isotopic signals in the North Pole region.

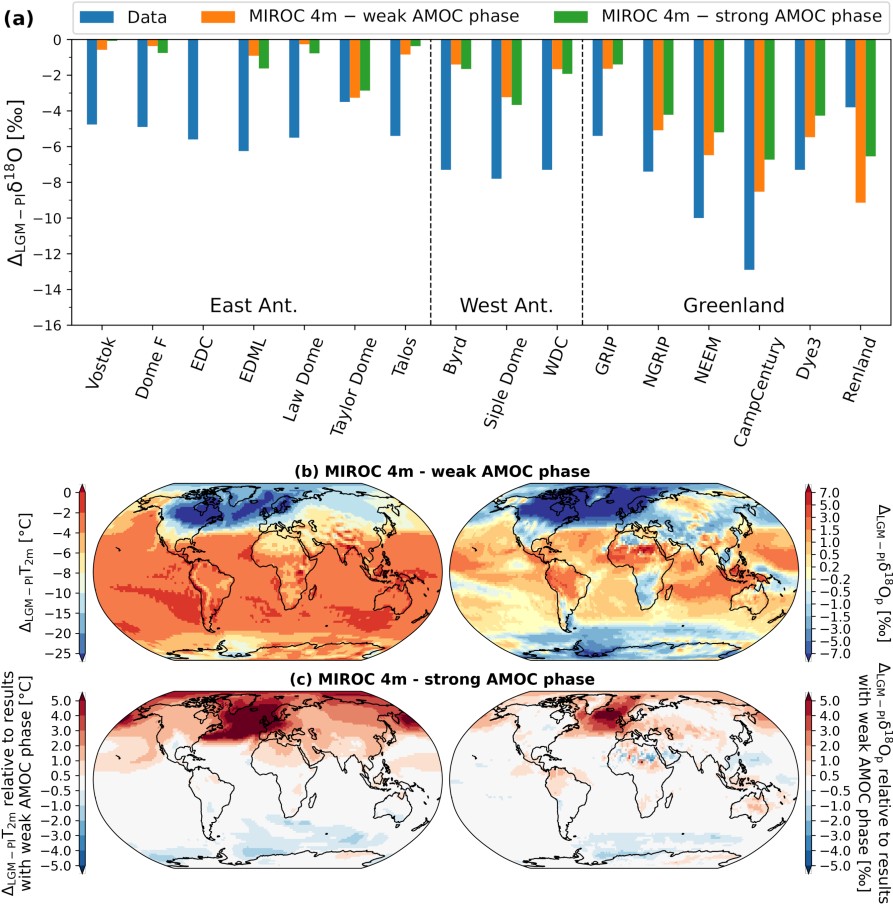

**Figure 7: (a) Comparison of modeled anomalies in $\delta^{18}O_{sn}$ between LGM and PI with $\delta^{18}O$ anomalies measured in polar ice cores**
**(blue bars). Modeled results are from simulations using the sea surface boundary conditions from the MIROC 4m coupled**



**simulations: LGM_miroc4m_sst_and_sic and LGM_miroc4m_strong_AMOC_sst_and_sic in orange and green, respectively. (b) Modeled T$_{2m}$ and $\delta^{18}$O$_p$ changes between LGM and PI using MIROC 4m (weak AMOC phase) sea surface boundary conditions (left and right maps, respectively). Maps in (c) show the impacts on T$_{2m}$ and $\delta^{18}$O$_p$ anomalies using sea surface boundary conditions from MIROC 4m at strong LGM AMOC phase. Values are expressed relative to the modeled results from (b).**

## 4 Impacts of sea surface boundary conditions on $\delta^{18}$O- T$_{2m}$ temporal slope for LGM-PI climate change

We have analyzed the effects of LGM to PI changes in SST and sea ice distribution on modeled $\Delta_{LGM-PI}\delta^{18}$O of precipitation and snow in the polar regions, as well as the impacts of the LGM AMOC strength. Next, we investigate the repercussions on modeled $\delta^{18}$O$_p$- T$_{2m}$ temporal slopes. In other words, are T$_{2m}$ and $\delta^{18}$O signals in the polar regions influenced in the same way by LGM to PI changes in SST and sea ice distribution? A correction for the prescribed glacial seawater change of 1 ‰ has been applied to LGM $\delta^{18}$O values before temporal slope calculation, according to equation 1 of Stenni et al. (2010). $\delta^{18}$O$_p$ values in the polar regions might be biased by strong changes in the seasonality or intermittency of the precipitation rate (Sime et al., 2009; Kino et al., 2021). To take into account this effect, modeled T$_{2m}$ values were weighted by the modeled monthly mean precipitation rates for the calculation of $\delta^{18}$O$_p$-T$_{2m}$ slopes (see Cauquoin et al., 2019b). As in Cauquoin et al. (2019b), the calculation of temporal slopes was restricted to grid cells where simulated temperatures are below +20°C for both PI and LGM. Moreover, we selected only the grid cells showing an absolute LGM-PI T$_{2m}$ difference of at least of 0.5°C. As a comparative point, PI spatial $\delta^{18}$O$_p$-T$_{2m}$ slopes of 0.72 and 0.94 ‰ °C$^{-1}$ are modeled by ECHAM6-wiso in East and West Antarctic ice core stations, respectively (calculated by considering the 25 grid cells centered on each drill location, excluding the ocean grid points), consistent with the mean observed value of 0.8 ‰ °C$^{-1}$ (Masson-Delmotte et al., 2008) and previous modeling studies (Schmidt et al., 2007, Werner et al., 2018, Cauquoin et al., 2019b). For Greenland ice core stations, we find a modeled spatial slope of 0.71 ‰ °C$^{-1}$, also in agreement with previous model results (Schmidt et al., 2007, Cauquoin et al., 2019b).

### 4.1 Antarctica

The values of $\delta^{18}$O$_p$-T$_{2m}$ slope in East Antarctica are influenced in different ways by sea surface boundary conditions. LGM to PI changes in sea ice area fractions have a strong impact on the slopes in coastal regions, as shown by the comparison between the plots (c)-(d) and (e)-(f) of Figure 8. Law Dome ice core is representative of this impact, with a slope of 0.29 and 0.62 ‰ °C$^{-1}$ depending if MIROC 4m (LGM_miroc4m_sst_and_sic, Figure 8d) or GLOMAP (LGM_miroc4m_sst_glomap_sic, Figure 8c) sea ice is used, respectively. The change of sea ice forcing has only a small effect on the temporal slopes modeled by ECHAM6-wiso in the East Antarctic plateau. The most sensitive case is EDC where the temporal slope is increased from 0.19 to 0.3 ‰ °C$^{-1}$ when switching from MIROC 4m sea ice with a strong AMOC to the GLOMAP one (Figures 8f and 8e, respectively). On average, the modeled temporal $\delta^{18}$O$_p$-T$_{2m}$ slopes of East Antarctic ice core stations are increased by more than 25% when MIROC 4m sea ice (red and brown markers in Figure 10) is replaced by the GLOMAP one (green and purple markers in Figure 10) due mainly to the coastal stations (i.e., Law Dome, Taylor Dome, Talos Dome and, to a lesser extent,





EDML). The conclusions remain the same if instead of taking the averages of the slopes at ice cores stations, we use the average slope across the entire East Antarctic area (Figure S6). The SST forcings have various impacts on the temporal slopes

simulated by ECHAM6-wiso. The SST forcing from Tierney et al. (2020) enhances the LGM cooling in the eastern part of the Southern Ocean area compared to other SST forcings (left map of Figure 5c). It influences both the LGM $T_{2m}$ and $\delta^{18}O_p$ in the same direction (i.e., toward lower values) but with different magnitudes at EDC, Vostok and Talos Dome. Temporal slopes at these stations are increased by 0.17, 0.11 and 0.08 ‰ °C$^{-1}$, respectively, when ECHAM6-wiso is forced by SST from Tierney et al. (2020) instead of the one from GLOMAP. The higher cooling in the Atlantic sector of the Southern Ocean when

ECHAM6-wiso is forced by GLOMAP SST (Figure 8a) makes the Antarctic temporal slope values higher between 0 and 90°E of longitude compared to the other simulations. It impacts especially the Dome Fuji and EDML ice core sites, where values of temporal $\delta^{18}O_p$-$T_{2m}$ slopes reach 0.8 and 0.67 ‰ °C$^{-1}$ (i.e., an increase of at least 60 and 34 %, respectively, compared to the other simulations). As a consequence, the modeled $\delta^{18}O_p$-$T_{2m}$ slopes in East Antarctic ice core stations are on average higher with GLOMAP SST forcing (blue marker in Figure 10). If all the East Antarctic area is considered, the forcing by the SST

from GLOMAP increases the average $\delta^{18}O_p$-$T_{2m}$ slope by more than 20% compared to the other SST fields (Figure S6).
Like in East Antarctica, a more extensive sea ice during LGM (i.e., GLOMAP) generally increases the modeled $\delta^{18}O_p$-$T_{2m}$ temporal slopes in West Antarctica. Except for the LGM simulation forced by Tierney et al. (2020) SST, the averages of temporal slopes for western Antarctic ice core stations are between 0.52 and 0.56 ‰ °C$^{-1}$ with GLOMAP sea ice, while they are between 0.45 and 0.52 ‰ °C$^{-1}$ if other sea ice forcings are used (Figure 10). This effect is larger in the Antarctic peninsula

and on the coast of the Amundsen Sea (Figures 8 and 11), influencing the average slope values in the entire western part of the continent (Figure S6). The use of Tierney et al. (2020) SST instead of the GLOMAP one (Figures 8b and 8a, respectively) as forcing for ECHAM6-wiso makes the cooling in the extreme western part of the Southern Ocean and of the Antarctic continent higher by 1 to 4°C (left map of Figure 5c) but enhances the $\delta^{18}O_p$ anomalies by only 3 ‰ at maximum (right map of Figure 5c). Therefore, the $\delta^{18}O_p$-$T_{2m}$ temporal slopes in Western Antarctica are on average decreased when ECHAM6-wiso is

forced with SST from Tierney et al. (2020) instead of GLOMAP (orange and blue markers in Figures 10 and S6, summarized in Figure 11).

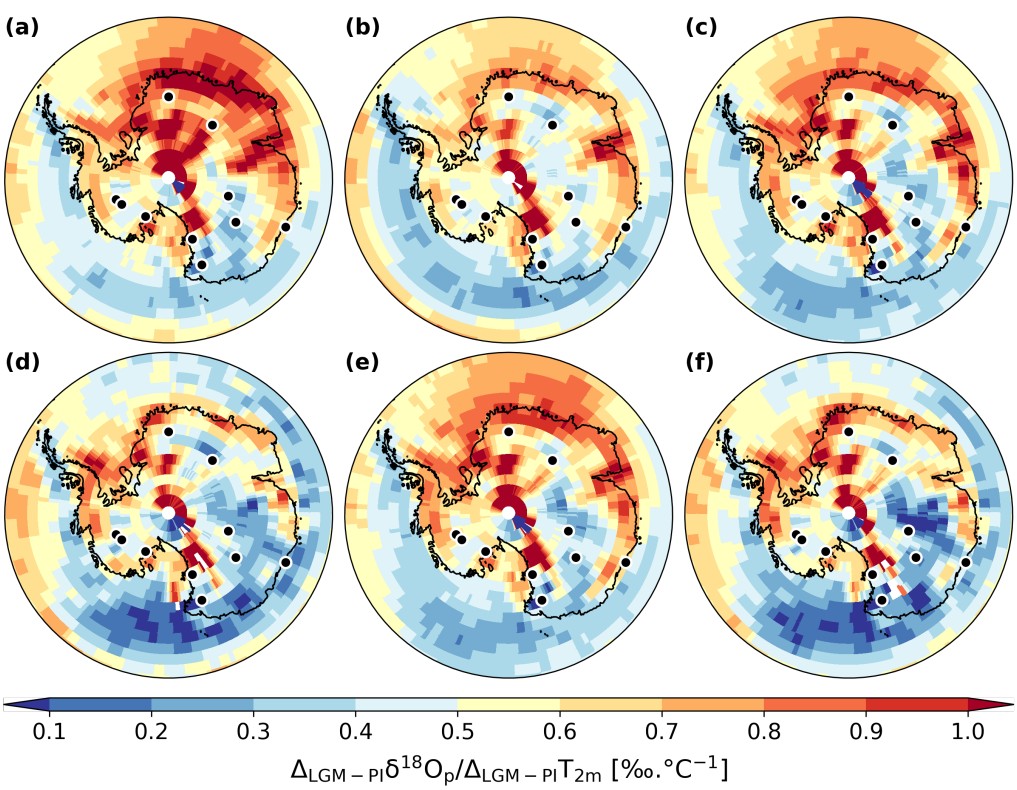

**Figure 8: Spatial distribution of $\delta^{18}O_p$-$T_{2m}$ temporal slope in Antarctica for LGM-PI changes according to our different ECHAM6-wiso simulations: (a) LGM_GLOMAP, (b) LGM_tierney2020, (c) LGM_miroc4m_sst_glomap_sic, (d) LGM_miroc4m_sst_and_sic, (e) LGM_miroc4m_strong_AMOC_sst_glomap_sic, and (f) LGM_miroc4m_strong_AMOC_sst_and_sic. The dots indicate the location of the ice core stations.**

## 4.2 Greenland

Using both SST and sea ice fields from GLOMAP as forcing for ECHAM6-wiso, we model higher $\delta^{18}O_p$-$T_{2m}$ temporal slope values at all Greenland ice core stations (Figure 9a) compared to all other simulations. This is generally true on the entire continent, too (blue marker in Figure S6). The average of the temporal slope values at ice core stations is 0.62 ‰ °C$^{-1}$ with GLOMAP sea surface boundary forcing (blue marker in Figure 10), and less than 0.52 ‰ °C$^{-1}$ in the other ECHAM6-wiso simulations. The influence of both SST and sea ice boundary fields explains this result.

The LGM-PI SST anomalies off the coast of Greenland are larger in GLOMAP compared to the other reconstructed and modeled SST fields (section 2.2.1), enhancing the cooling over North Atlantic Ocean (left maps of Figure 5) and so the LGM-PI anomalies in $\delta^{18}O_p$ (maps on the right side of Figure 5). The water vapor from this region is transported further north over the Greenland Sea during summer when sea ice shrinks. In Greenland Sea, local SST change is small while $\delta^{18}O_p$ anomalies are strong because of this water vapor transport. Then, the use of GLOMAP SST to force ECHAM6-wiso results in less cooling in the Greenland Sea area but stronger $\delta^{18}O_p$ anomalies compared to ECHAM6-wiso simulations using other SST boundary





conditions (Figure 5). As a consequence, the modeled $\delta^{18}O_P$-$T_{2m}$ temporal slopes are higher than 1 ‰ °C$^{-1}$ over the Greenland

Sea with GLOMAP SST (Figure 9a). The modeled slopes range only between 0.4 and 0.8 ‰ °C$^{-1}$ in the other simulations (Figures 9b to 9f). This affects the temporal slope at the Renland coastal station, where a temporal slope of 0.81 ‰ °C$^{-1}$ is simulated with GLOMAP SST, while this slope is below 0.6 ‰ °C$^{-1}$ in other simulations. Moreover, the larger cooling off the coast of Greenland with GLOMAP SST influences the modeled $\delta^{18}O_P$-$T_{2m}$ temporal slopes in inland Greenland ice core stations (Figure 9a) through changes in inland temperature (left maps of Figure 5) and also the inland transport of oceanic water vapor

from the North Atlantic Ocean and the Baffin Bay.

The use of GLOMAP or MIROC 4m sea ice boundary as forcing for ECHAM6-wiso simulations lead to mixed results in terms of modeled $\delta^{18}O_P$-$T_{2m}$ temporal slopes. The MIROC 4m sea ice in Greenland Sea shrinks less in summer compared to the one from GLOMAP (Figure 2). The effect on temperature is low but it enhances the LGM-PI anomalies in isotopic composition of precipitation over this area (Figure 6c), increasing the modeled $\delta^{18}O_P$-$T_{2m}$ temporal slopes (Figures 9d and 9f). It has a slight

effect on modeled temporal slopes (~0.1 ‰ °C$^{-1}$) over the eastern coastal regions of Greenland, including the Renland station. In ECHAM6-wiso, the isotopic composition of sea ice surfaces also reflects the isotopic composition of snow deposited on this surface. Then the formation of new sea ice from PI to LGM acts as a positive feedback effect in the decrease of surrounding $\delta^{18}O_P$, leading to steeper modeled $\delta^{18}O_P$-$T_{2m}$ temporal slopes (see text in Supplementary Material and Figure S7). Finally, ECHAM6-wiso forced with MIROC 4m sea ice, whose fractional areas are artificially lower (i.e., not 100% sea ice covered)

in coastal grid cells, simulates lower $\delta^{18}O_P$-$T_{2m}$ temporal slope values over Baffin Bay (between 0.3 and 0.6 ‰ °C$^{-1}$, Figures 9d and 9f) compared to when the model is forced with GLOMAP sea ice (between 0.7 and 1 ‰ °C$^{-1}$, Figures 9c and 9e). If the MIROC 4m sea ice is corrected by setting sea ice fraction as 100‰ as in GLOMAP (see text in Supplementary Material and Figure S8), we obtain temporal slope values similar to those in the simulations forced by GLOMAP sea ice (Figure S9). It also slightly increases the $\delta^{18}O_P$-$T_{2m}$ temporal slopes of inland ice core stations like NGRIP (0.53 and 0.67 ‰ °C$^{-1}$ with original

(Figure 9d) and modified MIROC 4m sea ice, respectively). This result shows that the presence or absence of sea ice very near the coast has a large influence on the modeled temporal slopes in some Greenland ice core stations (Figure 11).

The AMOC strength during LGM influences both the SST and the sea ice distribution in the Arctic region. While stronger LGM AMOC weakens the isotopic model-data agreement in Greenland because the predominantly less extensive sea ice reduces the modeled surface cooling (see section 3.4), it generally does not impact the temporal slopes modeled by ECHAM6-

wiso (red and brown lines in Figure 10) over inner Greenland. The ice core station that is most sensitive to the change in LGM AMOC strength is Renland, where the modeled $\delta^{18}O_P$-$T_{2m}$ temporal slope is decreased from 0.60 to 0.55 ‰ °C$^{-1}$ (Figures 9d and 9f).





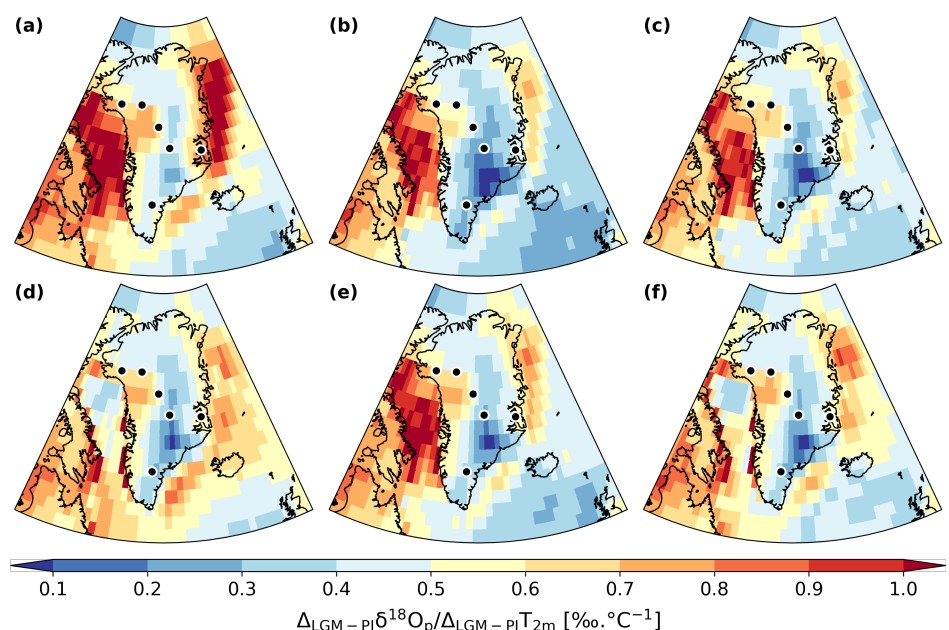

**Figure 9: Same as Figure 8 but for the Greenland region.**

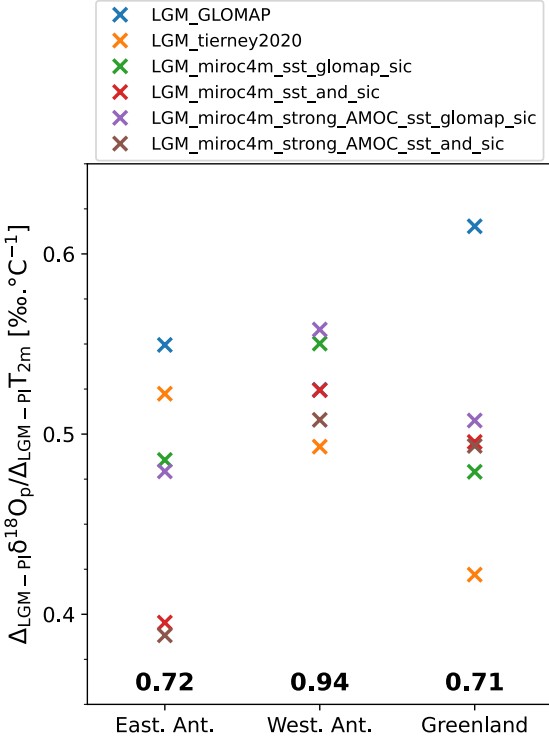

**Figure 10: Average modeled values of δ¹⁸O$_p$-T$_{2m}$ temporal slope for East Antarctic, West Antarctic and Greenland ice core stations according to our different simulations. Numbers in bold are the values of the corresponding modeled mean PI spatial slopes.**






## 5 Conclusions and Perspectives

In this study, we raised the importance of sea surface boundary conditions on the relationship between near surface air
temperature and $\delta^{18}O_p$ for LGM to PI climate change. Figure 11 illustrates the main findings of our study. In East Antarctica,
we noted the contrast between coastal regions and inland area in terms of control on the $\delta^{18}O_p$-$T_{2m}$ temporal slopes (left map
of Figure 11). The coastal site Law Dome is greatly affected by the LGM sea ice extent, with more than double the value of
the temporal slope and a better isotopic model-data agreement if GLOMAP sea ice is used instead of the MIROC 4m one
(Figures 8c and 8d, respectively). On the other hand, no noteworthy change in the $\delta^{18}O_p$-$T_{2m}$ temporal slope is simulated
regardless of the SST boundary conditions. A cooling larger by 1-2 °C in the Southern Ocean near Law Dome (Figure 5c)
changes the temporal slope value at this station only by a small amount (less than 0.05 ‰.°C$^{-1}$). The values of $\delta^{18}O_p$-$T_{2m}$
temporal slopes in inland ice core stations like Dome F, EDML, EDC, Vostok and Talos are mainly controlled by the change
of SST in our ECHAM6-wiso simulations. Stronger cooling in Atlantic sector of the Southern Ocean (GLOMAP) leads to
steeper temporal slopes in Dome F and EDML (between 0 and 40° E). Similarly, stronger cooling in the eastern part of the
Southern Ocean (Tierney et al., 2020) increases significantly the $\delta^{18}O_p$-$T_{2m}$ temporal slopes at EDC, Vostok and Talos. The
steeper $\delta^{18}O_p$-$T_{2m}$ temporal slopes are accompanied by better isotopic model-data agreement at these ice core stations (Figure
5a). The sea ice distribution can impact the $\delta^{18}O_p$-$T_{2m}$ temporal slopes at some inland stations, like EDML, but to a lesser
extent compared to coastal regions. In West Antarctica, we showed that sea surface boundary conditions have a mixed effect
on the $\delta^{18}O_p$-$T_{2m}$ temporal slopes (left map of Figure 11). For example, a steeper slope at WDC is simulated for more extensive
sea ice in the western part of the Southern Ocean, while no significant effect on the $\delta^{18}O_p$-$T_{2m}$ temporal slope at Byrd, located
very near WDC station, is seen. We note that a larger change in sea ice extent increases the mean $\delta^{18}O_p$-$T_{2m}$ temporal slope
across the entire West Antarctic region (Figure S6). Larger sea surface cooling in the western part of the Southern Ocean
slightly increases the $\delta^{18}O_p$-$T_{2m}$ temporal slopes at Byrd and WDC stations, and reduces it at Siple Dome. The lower change
in $\delta^{18}O_p$ compared to temperature could be explained by the influence of water vapor transport in this region. In Greenland,
our modeled results demonstrate clearly that the $\delta^{18}O_p$-$T_{2m}$ temporal slopes in Greenland are influenced by the sea surface
temperatures very near the coasts. The greater the LGM cooling off the coast of southeast Greenland, the larger the $\delta^{18}O_p$-$T_{2m}$
temporal slopes (right map of Figure 11). The cooling in this region of the North Atlantic in the GLOMAP reconstruction is
larger than in the Tierney et al. reconstruction or MIROC 4m model results, giving steeper modeled temporal slopes (0.62 and
0.42 ‰ °C$^{-1}$, respectively, on average on all Greenland stations) and generally a better agreement with isotopic data (Figure
5a). Similarly, the presence or absence of sea ice very near the coast can impact the modeled temporal slopes in some Greenland
ice core stations, and has a large influence in Baffin Bay and the Greenland Sea. The large southern extent of the sea ice is not
so important, as shown by the similar modeled temporal slope values using GLOMAP or MIROC 4m sea ice (weak AMOC
phase). On the other hand, the seasonal variation of the sea ice distribution, especially its retreat in summer, influences the
water vapor transport in this region and the modeled $\delta^{18}O_p$-$T_{2m}$ temporal slopes in the eastern part of the inner Greenland (right





map of Figure 11). Finally, while stronger LGM AMOC reduces the isotopic model-data agreement, it generally does not

impact the temporal slopes modeled by ECHAM6-wiso over Greenland. Variations in the temporal slope values are located in

the Greenland Sea, where the changes in surface temperature and sea ice distribution due to the AMOC strength mainly occur.

For Antarctica, only small changes in surface temperature and $\delta^{18}O$ are modeled by ECHAM6-wiso because the strong phase

period was selected in the middle of the AMOC peak. The impact of the period selection for the strong AMOC phase, like the

start or the end of the interstadial, will be investigated more in detail in a future study.

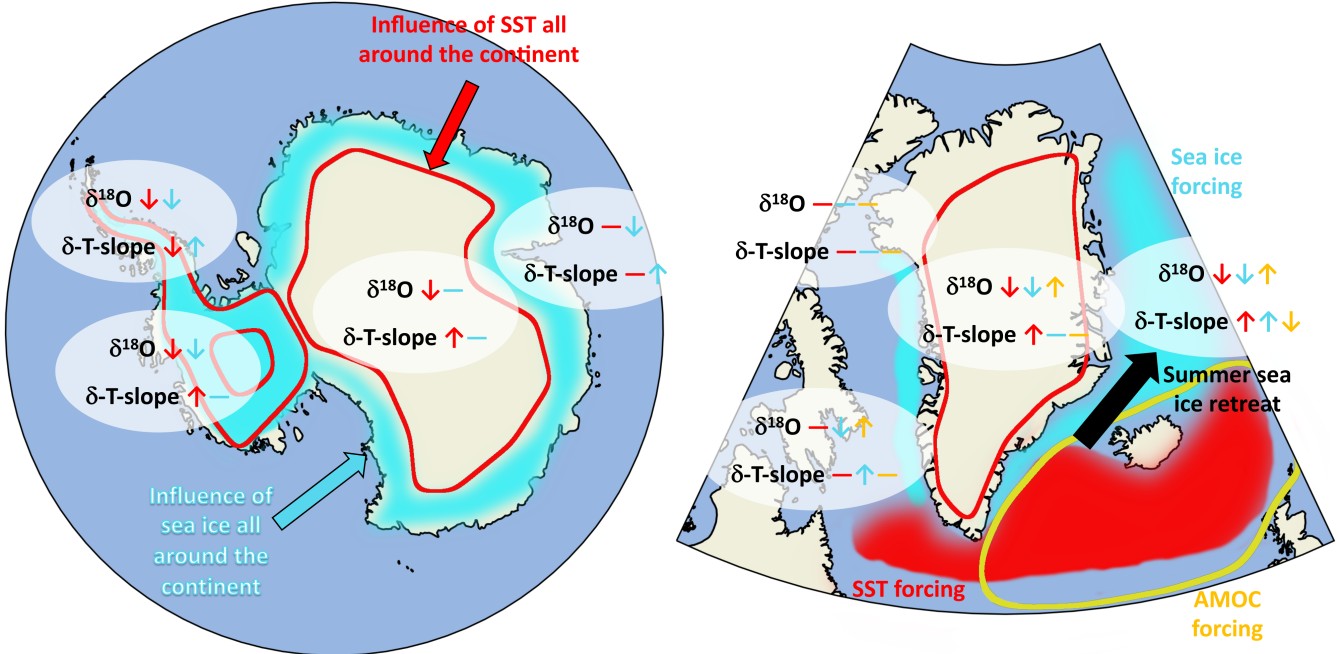

**Figure 11: Summary figure illustrating the influence of higher sea surface cooling, larger sea ice extent and stronger LGM AMOC (in red, cyan and yellow, respectively) on the modeled LGM $\delta^{18}O$ and temporal $\delta^{18}O_p$-$T_{2m}$ slopes in the Antarctic and Greenland regions (left and right, respectively). The up and down arrows indicate higher and lower values, respectively. The horizontal lines**
**indicate no significant change.**

In Greenland, ECHAM6-wiso simulates on average $\delta^{18}O_p$-$T_{2m}$ temporal slopes lower than the spatial one (0.71 ‰ °C$^{-1}$, Figure

10), as already reported in previous studies (Buizert et al., 2014; Cauquoin et al., 2019b; Jouzel et al., 1999; Werner et al.,

2000). In Antarctica, the ECHAM6-wiso modeled $\delta^{18}O_p$-$T_{2m}$ temporal slopes for LGM-to-PI climate change are on average

lower than the PI spatial slopes of the same model by at least 0.15 and 0.38 ‰ °C$^{-1}$ for eastern and western ice core locations,

respectively (Figure 10), regardless the simulation considered. By extension, we found much lower temporal slope values than

the ones estimated by Buizert et al. (2021). We simulate a maximum temporal slope value of 0.8 ‰ °C$^{-1}$ for Dome Fuji in the

LGM_GLOMAP simulation, while Buizert et al. (2021) found temporal slopes in Antarctic ice core stations ranging from 0.9

to 1.4 ‰ °C$^{-1}$, which are higher than the observed spatial $\delta^{18}O_p$-$T_{2m}$ slope of 0.8 ‰ °C$^{-1}$ (Masson-Delmotte et al., 2008).

Compared to PMIP3 ice sheet reconstruction, the use of GLAC-1D to run LGM simulations reduces the isotopic model-data

agreement for Antarctica (Figure S4). Also, the use of the old PMIP3 ice sheet reconstruction in ECHAM6-wiso increases the





resulting modeled $\delta^{18}O_P$-$T_{2m}$ temporal slopes (Figure S10) with mean values for East and West Antarctic ice core stations equal to 0.68 and 0.92 ‰ °C$^{-1}$, respectively, which are closer to the modeled mean PI spatial slopes (0.71 and 0.94 ‰ °C$^{-1}$, respectively) but still lower than the Buizert et al. (2021) results. So, the variability in LGM ice sheet reconstructions affects our modeled $\delta^{18}O_P$-$T_{2m}$ temporal slopes for LGM-to-PI climate change in Antarctica, as already shown by Werner et al. (2018).

On the other hand, we insist that the purpose of our study was to investigate the relative effects of sea surface conditions and AMOC strength the links between $\delta^{18}O_P$ and near surface air temperature, regardless the agreement or disagreement with other slope reconstructions.

In addition to orography effects, fractionation during the sublimation of surface ice is not taken into account in ECHAM6-wiso as in many isotope-enabled AGCMs. This process would lead to a further decrease in the $\delta^{18}O$ of water vapor in the polar
regions, contributing to steeper modeled $\delta^{18}O_P$-$T_{2m}$ temporal slopes in regions with low temperature. The mismatch between our model slopes and the reconstructed ones from Buizert et al. (2021) could be related to the representation of the atmospheric boundary layer and the related inversion temperature (Krinner et al., 1997; Masson-Delmotte et al., 2006; Cauquoin et al., 2019a), too. Still, despite these biases that potentially affect our modeled $\delta^{18}O_P$-$T_{2m}$ temporal slopes for LGM-to-PI climate change, our ensemble of simulations provides information on how sea surface conditions partially control the links between
$\delta^{18}O_P$ and near surface air temperature in polar regions.

Because only ECHAM6-wiso is used in this study, we cannot exclude the model-dependency of our results. So, the use of isotope-enabled AGCMs other than ECHAM6-wiso would be beneficial to confirm or refute our findings. A set of SST reconstructions for the LGM, based on both model results and observations, are now available. We raise the importance of providing sea ice cover reconstruction for this period too. The sea ice cover simulated by coupled GCMs for the LGM period
takes various forms. An alternative reconstruction to the GLOMAP one, also based on observations, would help to better assess the impact of sea ice cover on the $\delta^{18}O_P$-$T_{2m}$ relationship for LGM to PI climate change. As a first step, the focus of this study was to identify and quantify the important factors influencing the isotope-temperature relationship in the polar areas for the LGM to PI climate change. Future studies will investigate the evolution of this relationship along the whole of the last deglaciation. For that, an ensemble of equilibrium isotopic simulations using the sea surface and ice sheet boundary conditions
from MIROC 4m for different time periods between the LGM and PI will be performed.

*Code availability.* The ECHAM model code is available under a version of the MPI-M software license agreement (https://www.mpimet.mpg.de/en/science/models/license/). The code of the isotopic version ECHAM6-wiso is available upon request on the AWI's GitLab repository (https://gitlab.awi.de/mwerner/mpi-esm-wiso).


*Author contributions.* AC designed the model experiments and performed the simulations using the MIROC 4m sea surface boundary conditions with the help of AAO, TO and WLC. AC performed the simulations using the GLOMAP or Tierney et



al. sea surface boundary conditions with the help of MW and AP. AC and all the co-authors analyzed the model outputs. AC wrote the manuscript with contributions from all co-authors.


*Competing interests.* One of the co-author (André Paul) is editor in Climate of the Past.

*Acknowledgements.* This research was supported by JSPS KAKENHI Grant (17H06323) and by the German Federal Ministry of Education and Research (BMBF) as Research for Sustainability initiative (FONA). The ECHAM6-wiso simulations were
performed at the Alfred Wegener Institute (AWI) supercomputing center. The MIROC 4m simulation used in this study was performed on the Earth Simulator 3 at Japan Agency for Marine-Earth Science and Technology (JAMSTEC).

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
