# Peer review of "Effects of LGM sea surface temperature and sea ice extent on the isotope-temperature slope at polar ice core sites"

_Climate of the Past, 2023_

## Author Comment (AC1)

**Response to the comments of Reviewer 1**

I went through the manuscript by Cauquoin et al. titled *"Effects of LGM sea surface temperature and sea ice extent on the isotope-temperature slope at polar ice core sites"*. The study investigates how SST and sea ice cover have an impact on the spatio-temporal variability of the $\delta^{18}O$ vs T slope for Greenland and Antarctica precipitation between LGM and preindustrial climate. The authors used the ECHAM6-wiso model forced with different boundary conditions to test the impact of reconstructed and modeled SST/sea ice cover on simulation output. The key result of the work is that $\delta^{18}O$ vs T slope is modulated by combination of both forcing (SST and sea ice, plus AMOC for Greenland) with different weights that depend on geographical location. The authors also highlight the importance of using reconstructed sea surface boundary conditions instead of using coupled models output and specifically the needing of sea ice cover reconstruction for LGM period.

**General comment**

This work provides an important piece of information to the isotope - glaciology community, because it shows that **(1)** SST and sea ice conditions over source regions of precipitation have an impact on the reconstructed temperatures using stable isotopes in ice cores and **(2)** the impact on the isotope-temperature temporal slope is location-dependent over the two continents. In this context, Figure 11 clearly show where such driving forces affect more the slope and the $\delta^{18}O$ of precipitation. In my opinion, the manuscript is highly relevant for CP audience, is well written, and is easy to read. Therefore, I strongly support the manuscript for publication and I have only minor-technical comments reported hereafter:

We thank the reviewer 1 for his/her appreciation of our paper.

**L225-227** and **Figure 4**. A metric to evaluate the agreement could be useful (e.g. correlation or RMSE), similar to the metrics reported in table 3 for the slope.

We added the values of slope and RMSE in the plot (d) of Figure 4. These values are reported in the Table 3 of the initial manuscript too (Figure 4 corresponds to the simulation LGM_miroc4m_sst_and_sic).

[Figure]

**L235** This sentence is a bit vague. Are the authors referring to the spatial-temporal distribution of $\Delta_{LGM}-PI\delta^{18}O_{P}$? Or is this a "general" sentence? In that case, I would replace the word *distribution* with *fractionation*.

We replaced the word "distribution" with "fractionation" (l. 255).

**L363** The scientific question guiding section 4 is very clear and it should be also posed in the introduction.

Done (l. 90-94): "Are air temperatures near the surface and the isotopic composition of precipitation in the polar regions influenced by LGM to PI changes in SST and sea ice distribution in the same way? What are the underlying dynamics, for example, in terms of changes in concentrations and transport of water vapor? To answer to these questions, we performed…"

---

## Author Comment (AC2)

**Response to the comments of Reviewer 2**

Cauquoin et al. present a series of LGM simulations using the ECHAM6-wiso isotope-enabled atmosphere GCM, and use these to investigate the influence of SST, sea ice extent, and AMOC on the LGM isotopic depletion and isotope-temperature slopes in Greenland and Antarctica. While the work appears to be free from major technical errors, the reader unfortunately does not learn much from the lengthy study beyond what is known from earlier work – other than perhaps the fact that isotopic slopes are complicated. This is mostly due to an experimental design that is not ideal to discern the effects that the authors seek to study. Another problem is the interpretation of the data that focuses mostly on lengthy anecdotal descriptions of the simulation outcomes rather than an analysis of underlying dynamics.

Since it is too late to adjust the experimental design, it seems the paper should probably be published. However, I would request the authors consider the changes suggested below.

We thank the reviewer 2 for his/her useful suggestions that helped to improve substantially our manuscript. We tried to facilitate the reading of our paper by referring less to the name of sea surface boundary conditions or the simulations' name and more to their characteristics (stronger cooling, more extensive sea ice...). Moreover, we added two figures in the main text and 2 other ones in supplementary material to analyse the changes in moisture dynamics influencing the modeled temporal slopes.

**Major comments:**

Comment 1:
First I want to share my concerns with the experimental design. This design does not allow for easy or straightforward assessments of impacts of SST and SIC. The various forcing files used are pulled from independent sources, and therefore have strong spatial (and presumably seasonal) differences between them that complicate the interpretation. For example, the effect of SST is mostly found by comparing GLOMAP, Tierney et al., and MIROC. The differences between them have strong spatial patterns such that some parts of the Southern Ocean are colder in one, and other parts colder in the other. This makes the interpretation of the SST influence very complicated. It would have been much simpler had the authors decided for example to simply subtract or add 1 degree to the entire GLOMAP SST anomaly – or perhaps scale the anomaly by a constant value. The same holds for the sea ice concentrations, where simply applying SIC anomalies would have been much more insightful. Another downside of their approach is that the reader has to do a lot of work to understand the manuscript, which is dense with acronyms. Is the GLOMAP SST colder than the MIROC_4m_strong_AMOC, and in which parts of the ocean? Which of the three SIC files is most extensive? The manuscript shows and explains this, but the average reader will not be able to keep all these facts straight in their minds while reading the manuscript and interpreting the figures. I found myself having to go back and forth all the time to understand what is being discussed which adds a lot of reading time to already (unnecessarily) long paper.

Answer to comment 1:
We agree with the reviewer that the differences between the different SST boundary conditions can complexify the interpretation. It is not that true for the sea ice, though, where simply applying sea ice anomalies is not so straightforward.

We also agree that to keep in mind the differences between the GLOMAP SST and MIROC 4m one for example can be complicated. We corrected the manuscript by being more concrete in the description of the differences, without necessarily referring to the name of the SST/sea ice conditions but more to their characteristic (colder, more extensive…). For example in section 3.2, the old sentence

"However, a strong cooling is obtained with GLOMAP SST in the Southern Ocean, which is the evaporative source of isotopic signals measured in polar areas. As a consequence, temperature changes in Antarctica are stronger when using SST from GLOMAP or Tierney et al., giving higher modeled $\delta^{18}O_p$ changes compared to modeled results using SST fields from MIROC 4m (right maps of Figure 5), and better agreement with the observations."

was changed to

"For a stronger SST cooling in the Southern Ocean (GLOMAP and Tierney et al.), ECHAM6-wiso simulates higher $\delta^{18}O_p$ changes (right maps of Figure 5) that are in better agreement with the observations."

Comment 2:

The authors use an incorrect metric of temperature in all their slope analyses. They use the precipitation-weighted T2m temperature (line 367). The long-standing standard in the literature is to use the annual mean temperature in estimating isotope slopes, in part because the precipitation-weighted temperature is never known in observational situations. For example, the spatial slope of 0.8 permil/K that the authors cite and compare to is based on mean-annual temperatures (Masson-Delmotte et al., 2008). It is widely understood that the small temporal isotope slopes in Greenland of around 0.3-0.4 permil/K are due to a change in precipitation seasonality, which is not expected in Antarctica. The metric by the authors would not reflect this difference. The temporal slopes the authors present can therefore not be meaningfully compared to values found elsewhere in the literature. While others have used precipitation-weighted temperature (besides mean annual) as part of an analysis to understand isotope dynamics, it should not be used as the only estimate.

Answer to comment 2:

We agree that our modeled temporal slopes in the initial version of the manuscript could not be compared with the reconstructed ones. We use the annual mean temperature in the revised manuscript (Figures 8, 10, 12, S9, S10, S12 and S13 in the revised manuscript). Our initial thought was more to investigate the inter-simulations differences than to compare our modeled slopes to existing slope reconstructions. Indeed, we are aware that ECHAM6-wiso simulates too low temporal slopes in Antarctica. Despite that, it is still interesting to evaluate the impacts of SST and sea ice changes on these modeled slopes, regardless the agreement or disagreement with slope reconstructions.

**Comment 3:**

The manuscript is much too long, I believe. While CP does not have page limits, the readers (and reviewers!) would appreciate a much more concise manuscript. The lengthy descriptions of observations can be shortened, as the readers can glean the same from the figures.

**Answer to comment 3:**

We agree that some parts of the article were too descriptive somehow, especially in sections 3 and 4, giving the impression of a long text. We tried to shorten the section 3 and rewrote completely the section 4 that is now more focused on the explanation why we get such changes in temporal slopes through changes in the moisture transport (with 2 more figures, see answer to comment 5). We also shortened the section 2.4 (Observational data) by moving the technical aspects in Supplementary Material (Text S2). After revision, the manuscript is not so shorter but hopefully easier to read and more interesting in the section 4.

**Comment 4:**

The authors should perform more meaningful model-data comparison. Currently they only compare the change in d18O. For both Greenland and Antarctica empirical temperature reconstructions exist from e.g. borehole reconstructions, which give direct estimates of past temperatures and the temporal isotopic slope. A model-data comparison of change in T and of temporal isotope slopes would be very insightful, and allow the reader to judge whether the model has skill. For the d18O model comparison the model appears to underestimate the d18O changes – this would imply that the simulated isotopic slopes are likely biased toward too small values, but this is not discussed or shown. Figure 10 shows no significant differences between isotopic temporal slopes in Greenland and Antarctica, while it is a well-established observational fact that temporal slopes in Greenland are smaller.

**Answer to comment 4:**

We added a comparison to temperature reconstructions in the section 3 of our manuscript. See below the new figure 6 as example (LGM-PI temperature changes indicated by colored markers in plot a, and the related temperature scale added on the right y-axis). While ECHAM6-wiso is generally biased toward too high $\Delta\delta^{18}O$ values in Antarctica (LGM-PI changes in $\delta^{18}O$ are not strong enough), we can see that the model reproduces generally well the observed Antarctic temperatures. As the reviewer noted, it implies too small isotope-temperature temporal values in Antarctica.

[Figure]

For the temporal slopes, we re-wrote the paragraph in the conclusion section 5 (l. 653-663):
"In Greenland, ECHAM6-wiso simulates $\delta^{18}O_p$-$T_{2m}$ temporal slopes oscillating between 0.2 and 0.7 ‰ °C$^{-1}$ inland and at northwestern coastal sites, respectively, lower than the spatial one (0.71 ‰ °C$^{-1}$, Figure 12), as already reported in previous studies (Buizert et al., 2014; Cauquoin et al., 2019b; Jouzel et al., 1999; Werner et al., 2000). Our modeled temporal slope values for stations NEEM (around 0.7 ‰ °C$^{-1}$) and NGRIP (between 0.37 and 0.57 ‰ °C$^{-1}$) are in agreement with previous reconstructions (Buizert et al., 2014), too. In Antarctica, the ECHAM6-wiso modeled $\delta^{18}O_p$-$T_{2m}$ temporal slopes for LGM-to-PI climate change are on average lower than the PI spatial slopes of the same model by at least 0.20 and 0.48 ‰ °C$^{-1}$ for eastern and western ice core locations, respectively (Figure 12), regardless of the simulation being considered. By extension, we found much lower temporal slope values than the ones estimated by Buizert et al. (2021). We simulate a maximum temporal slope value of 0.9 ‰ °C$^{-1}$ for the South Pole, while Buizert et al. (2021) found temporal slopes in Antarctic ice core stations ranging from 0.9 to 1.4 ‰ °C$^{-1}$, which are higher than the observed spatial $\delta^{18}O_p$-$T_{2m}$ slope of 0.8 ‰ °C$^{-1}$ (Masson-Delmotte et al., 2008)."

Indeed, the modeled temporal slopes in Antarctic ice cores stations are generally too low while those in Greenland are in good agreement with the reconstructed ones. It explains why there is no huge difference between temporal slope in Antarctica and Greenland (new Figure 12, see left panel below). When considering the entire land area for East Antarctica, West Antarctica and Greenland, a clearer contrast can be seen between south and north polar regions (new Figure S9, see the right panel below).

[Figure]

Note for these figures: the colors of the markers have been revised to consider color vision deficiencies.

Comment 5:

The authors seek to capture the influence of SIC and SST on isotopes at the various sites in their Fig. 11, but do not provide any insight into why the patterns are the way they are. Why does SIC impact coastal but not inland sites? Modern isotope-enabled models have a suite of tools to address such questions, such as for example moisture tagging. Can you provide more insight? Noone and Simmonds (2004) already provided a very thorough interpretation on the SIC impact on d18O, and I would expect a follow-up study to provide more or deeper insight - which is lacking here. There is no meaningful attempt to understand or analyze the atmospheric dynamics or moisture transport.

Answer to comment 5:

ECHAM6-wiso is not equipped with water tagging, yet. However, we re-wrote completely the section 4 to focus on the analyses of the temporal slope changes from a moisture transport point of view (and hopefully in a less lengthy descriptive way).

[Figure]

The figure above (Figure 9 in the revised manuscript) shows the anomalies in vertically integrated water vapor transport (arrows) and integrated column of water vapor (colored backgrounds) for (b) more cooling in the Southern Ocean, (c) more extensive sea ice and (d) stronger AMOC.

With a stronger SST cooling in Southern Ocean, the westerlies around Antarctica are enhanced and the atmosphere is wetter in the mid-latitudes while a drier belt is around the continent (plot b). More water vapor is transported from the lower latitudes (30-40° S) of the Atlantic sector. As this water vapor is relatively depleted in $\delta^{18}O$ because of enhanced evaporation by 20 to 30% there (see Figure 5d), it increases the temporal slopes by at least 25% in EDML and DF compared to other simulations with less cooling in Southern Ocean. Also, we found that more SST cooling near the Amundsen Sea decreases the water vapor transport from this region to western Antarctica sites (new Figure S7 shown in the left map below). The $\delta^{18}O$ change of the water vapor from this source area is relatively less strong (2 ‰ at maximum) compared to the decrease in local temperature (2 to 4 °C). So, less contribution from this source region to West Antarctica inland increases the temporal slopes at WDC and Byrd stations. At the same time, this water vapor influences the $\delta^{18}O_p$ of nearby coastal region like the Antarctic peninsula, making decrease their temporal slopes.

[Figure]

A more extensive sea ice does not change drastically the transport of moisture but the nature of moisture origin (sublimation of snow on sea ice or evaporation of open water). It influences mainly coastal sites. On the other hand, a more extensive sea ice increases the slope in a geographical band area from Law Dome to Vostok and EDC stations. It is explained by a decrease of water vapor transport with higher $\delta^{18}O$ concentrations from the Indian Ocean and the south of Australia, especially in austral winter (new figure S8 shown in the right plot above).

[Figure]

(a) GLOMAP  (b) More Arct. Oce. SST cooling  (c) More extensive sea ice  (d) Stronger AMOC

For Greenland (figure above is figure 11 in the revised manuscript), more moisture from the Northern North Atlantic (US coast) are transported to Greenland and Arctic area when there is more cooling in the Arctic region. This water vapor depleted in $\delta^{18}O$ makes increase the $\delta^{18}O$/temperature temporal slope in Greenland.

A more extensive sea ice makes the Arctic Ocean area drier, especially at 50° N, and it slightly slows down the transport of water vapor from the North Atlantic to Greenland area (plot c). On the other hand, all this area is covered by sea ice in the "more extensive sea ice" simulation (i.e., MIROC 4m sea ice). It makes decrease the $\delta^{18}O$ of water vapor above this surface, increasing the isotope-temperature temporal slope in eastern Greenland coast and Greenland Sea. In the latter, a more extensive sea ice especially in summer makes decrease the LGM $\delta^{18}O$ too while the effect on temperature is low, increasing again the local temporal slope.

A stronger AMOC increases the amount of water vapor and enhances its transport from the North Atlantic to European coasts because of the less extensive sea ice (plot d). More water vapor with higher $\delta^{18}O$ are available at southeast of Greenland because of the substitution of sea ice by open water. However, there is only a slight increase in the transport of this water vapor toward north in Greenland interior (plot d) while the cooling inland is largely reduced (Figure 7c). So, isotope-temperature temporal slopes are slightly increased over inner Greenland for stronger AMOC (dark and light purple markers in Figures 12 and S9). On the contrary, temporal slopes are decreased over Greenland Sea because of the presence of open water instead of sea ice, enhancing locally the LGM $\delta^{18}O_p$.

These analyses are added in the summary figure 13 below.

[Figure]

Comment 6:

I think providing the reader with more understanding of the moisture sources of the various stations would be helpful for their understanding. It has long been known that low elevation/coastal cores derive more of their moisture form local nearby sources, whereas high-elevation cores receive their moisture from long-ranged transport (Sodemann & Stohl, 2009). Since sea ice impacts the regional waters around Antarctica, I would expect Antarctic sea ice to impact coastal stations and not inland ones, as the authors indeed find.

Answer to comment 6:

Thank you very much for this reference. We added information on the moisture sources of the polar stations in section 4:

Lines 459-460: "$\delta^{18}O$ in coastal and western low-elevated sites are derived from nearby local sources while the sources of high-elevation East Antarctic ice cores are typically further north around 40-45° S (Sodemann and Stohl, 2009)."

Lines 516-518: "For Greenland, most of the moisture comes from Northern North Atlantic Ocean at latitudes 30-40° N (Drumond et al., 2016), south of the ice sheet (Figure 11a)."

Comment 7:

The authors state several times that the PMIP3 ice sheet would improve the d18O simulations relative to the GLAC-1D ice sheet. However, most glaciologists would agree that PMIP3 is not the most realistic ice sheet (its elevation is much too high in the interior), and instead prefer GLAC-1D or ICE-6G. The PMIP3 ice sheet would lead to stronger Antarctic cooling via the higher elevation, and thereby deplete the isotopes and improve the ECHAM6-wiso fit to d18O observations. Do you think the PMIP3 ice sheet improves realism, or simply compensates for a model bias through anomalously high interior elevation? Please elaborate on your thinking.

Answer to comment 7:

The way how this aspect of ice sheet changes was presented was misleading. We do not say that the PMIP3 ice sheet is better than GLAC-1D, because it is not the case. Indeed, we do see an improvement in isotopic model-data agreement for Antarctic area when using PMIP3 ice sheet. But we cannot exclude that this is due to an isotopic bias in ECHAM6-wiso that is counter-balanced using a thicker ice sheet. Moreover, the use of this ice sheet degrades the temperature model-data agreement in Antarctica (crosses in Figure S5 below shown below). We added such statement in the manuscript.

Section 3.2 (l. 297-299): "However, this better $\Delta\delta^{18}O$ model-data agreement is more likely due to a bias compensation than a more realistic ice sheet because the simulation of Antarctic temperatures by ECHAM6-wiso is degraded at the same time (markers in Figure S5)."

Section 5 (l. 663-671): "The use of the thicker PMIP3 ice sheet reconstruction compared to GLAC-1D increases the resulting modeled $\delta^{18}O_p$-$T_{2m}$ temporal slopes in ECHAM6-wiso (Figure S13) with mean values for East and West Antarctic ice core stations equal to 0.68 and 0.92 ‰ °C$^{-1}$, respectively, by decreasing the isotopically enriched bias in the model for LGM (Figure S5). However, the temperature model-data agreement is reduced in this case."

[Figure]

**(a)**

Legend: Data (blue) | GLAC-1D LGM ice sheet (orange) | PMIP3 LGM ice sheet (green)

East Ant. | West Ant. | Greenland

x-axis labels: South Pole, Vostok, Dome F, EDC, EDML, Law Dome, Taylor Dome, Talos, Byrd, Siple Dome, WDC, GRIP, NGRIP, NEEM, Camp Century, Dye3, Renland

Left y-axis: $\Delta\delta^{18}O$ [‰]
Right y-axis: $\Delta T_{2m}$ [°C]

Comment 8:
Figure 8: Why is there such strong spatial variability in the isotopic slopes on such small spatial scales? Is this a model artifact? Would this go away with averaging over longer timescales? The ice core d18O observations are very consistent between cores, and temperature is likely to be homogeneous also.

Answer to comment 8:
The modeled values are already averaged over a relatively long timescale, so this is not the origin of the patchy pattern in central Antarctica. The original figure 8 shows the values of temporal slope in each grid cell. There could be some biases because the grid mesh is more and more tightened when going towards the poles. The use of the annual mean temperature instead of the precipitation-weighted one improved the situation. Moreover, to facilitate the readability of the figure, we drew filled contours instead of the colors in each grid cells (see below the figure 8 from the corrected manuscript).

[Figure]

$\Delta\delta^{18}O_p/\Delta T_{2m}$ [‰.°C$^{-1}$]

Comment 9:
It is unclear what the implications of the work are for others working in the field. Do the authors have any recommendations for the future interpretation of water isotopes?

Answer to comment 9:
We stress the importance of LGM sea ice boundary conditions on the $\delta^{18}O$ signals of Greenland and Antarctica, including some inland sites like EDC and Vostok (l. 686-688). The reconstruction of temperature in the western part of Antarctic continent is complexified by various effects of sea ice and SST (related to change of moisture transport and origin) depending on the considered sites (l. 619-623). Finally, we also emphasize that more proxy measurements of temperature and sea ice are necessary for the Southern Ocean. Relatively large uncertainties remain in the reconstruction of the climatology in this area while the water vapor from this region contributes largely to $\delta^{18}O_p$ in Antarctica. We added such statements in the conclusion section (l. 690-692).

**Line-by-line comments:**

Line 22: what are "mixed effects"?
We rephrased this sentence (l. 24-26): "Effects of sea surface boundary conditions changes on isotope-temperature temporal slopes are simulated in West Antarctica are various. This is due partly to the transport of water vapor from the Southern Ocean to this area that can dampen…"

Section 2: somewhere you should explain what AMIP is.
Done (section 2.3, l. 160-161): "The LGM SST boundary fields are expressed relative to the Atmospheric Model Intercomparison Project (AMIP, Eyring et al., 2016) mean SST…"

Section 2.2: Can you also specify the PI conditions you use?
We clarified the experimental design of our PI simulations (PMIP4 protocol) in section 2.3 Model setup and experiments.

Line 123: These simulations with different AMOC values are therefore not in equilibrium, correct?
As we say at lines 119-120 of the original manuscript, the MIROC 4m LGM simulation has an oscillating AMOC strength. However, the selected periods (weak and strong AMOC) are in quasi-equilibrium as you can see in the figure S1 (see answer to next question).

Line 127: What does this mean: "selected in the middle of the AMOC peak"?
It means that we selected a 100-year period from within the middle of the AMOC peak to create the SST and sea ice climatology fields for a strong AMOC. We could have selected instead the highest peak of AMOC for instance (at around 26 000 years in the graph below). As a picture is very often better than a written description, we added a new Figure S1 (see below) showing the AMOC variations in MIROC 4m simulation and the periods selected for strong and weak AMOC phase to build the corresponding SST and sea ice average boundary conditions.

[Figure]

Line 155: Can you use SST and SIC reconstructions that are not self-consistent? Are there risks, such as warm temps under sea ice or freezing SST conditions without sea ice?
In ECHAM6-wiso, the SST is modified according to the presence or absence of sea ice. If there is sea ice, the SST is set to the freezing temperature (i.e., 271.38 K). If there is no sea ice, the SST is set to the maximum value between the one provided in the SST input file and the freezing temperature. We added a statement at lines 165-167: "Since we also used GLOMAP sea ice extent data in this case, the SST was adjusted slightly to maintain consistency (e.g., SST set to freezing temperature where there is sea ice)."

Table 1: It would be more clear in the last column to state "Less SST cooling" and "more SST cooling"
Done ("Less global SST cooling" and "More global SST cooling").

Figure 3: why not add the South Pole ice core? Data are publicly available.
Done in all the relevant figures and Tables 2 and 3.

Line 246: You could consider removing the LGM-PI subscript. You compare the same periods throughout the paper, so it's unnecessary to specify all the time. It would improve readability.
Done in the text and figures.

Table 2: could you add the simulated range of d18O?
Done.

Line 221-222: "due to lower temperatures". Is this cause something you assume to be true, or tested somehow? Please specify
We rephrased this sentence (l. 238-240): "Generally, negative $\delta^{18}O_p$ anomalies are also simulated over Antarctica and the Southern Ocean, where the LGM cooling is stronger compared to lower southern latitudes (Figure 4a)."

Fig 4b: instead of plotting the change in P, would it make sense to plot the ratio (P_LGM/P_PI)?

Done (Figures 4 and S3).

Fig. 4d: all data are in the upper left half. Do you have thoughts why?
The sub-tropical ice cores are sensitive to changes in precipitation. A bias in tropical LGM-PI precipitation changes could explain the generally too weak modeled $\delta^{18}O$ changes. Another reason could be the relatively low-resolution topography in the model (1.9°), especially for the Himalayan and polar areas. Still for the polar regions, fractionation during the sublimation of surface ice is not taken into account in ECHAM6-wiso as in many isotope-enabled AGCMs. This process would lead to a further decrease in the $\delta^{18}O$ of water vapor in the polar regions, giving a better isotopic model-data agreement (and contributing to steeper modeled $\delta^{18}O_p$-$T_{2m}$ temporal slopes in regions with low temperature). Finally, the isotopically enriched bias could be also due to the representation of the atmospheric boundary layer and the related inversion temperature. These explanations are in the penultimate paragraph of the conclusion section.

Line 243: "strong cooling" – is this global, or just Arctic?
Just the Arctic region (l. 264): "For the Arctic region, a strong cooling is simulated with the very extended sea ice from MIROC 4m…"

Line 270: Why do you think this improves the data? Is the ice more realistic, or does it compensate a model bias? I would suspect the latter
The better model-data agreement in $\delta^{18}O$ is likely due to a model bias compensation, as shown by the degraded simulation of temperatures in Figure S4 (l. 297-299): "However, this better $\Delta\delta^{18}O$ model-data agreement is more likely due to a bias compensation than a more realistic ice sheet because the simulation of Antarctic temperatures by ECHAM6-wiso is degraded in the same time (markers in Figure S4)."

Line 340: I am surprised about the weak influence of the AMOC on Greenland climate. During DO events, Greenland warms by around 10 degrees during AMOC strengthening.
The influence is in the range of 2°C to 6°C. In MIROC 4m simulation, the difference between stadial (weak AMOC) and interstadial (strong AMOC) is about 5°C. However, this experiment is like a spin-up simulation with input conditions fixed to LGM. In other MIROC 4m experiments that are actually transient ones with values of greenhouse gases and orbital parameters varying realistically over a long period of time, the temperature difference is in the order of 10°C. If we would want to compare temperature changes across DO events, then we should use these transient experiments and not the LGM one.

Line 367: You should not use precip-weighted temperatures. Almost the entire literature in this field reports mean annual.
We use now the annual mean temperatures for the calculation of the temporal slopes (see answer to comment 2 and see introduction part of section 4 at lines 414-416): "…the calculation of temporal slopes was restricted to grid cells where simulated annual mean temperatures are below +20°C for both PI and LGM. Moreover, we selected only the grid cells showing an absolute LGM-PI annual mean $T_{2m}$ difference of at least of 0.5°C."

Line 373: The 0.8 permil/K is regressed against mean annual temperatures, so you cannot compare to your values

Our modeled PI spatial slopes for East and West Antarctica (0.72 and 0.94 ‰/K) were calculated with the annual mean temperatures, so they are comparable with the observed 0.8 ‰/K value.

Line 384: these values (slope around 0.25 permil/K) suggest a bias in the model, no? If real, this would correspond to 24K of LGM cooling at EDC.

Yes, there is an isotopically low bias in the LGM-PI $\delta^{18}O$ changes in Antarctica. See our response to the related main comment.

Fig. 8: Why are these maps so patchy?

See our response to comment 8.

Line 426: in *the* Greenland sea

This section has been completely re-written.

Line 436: lead*s* to mixed results

This section has been completely re-written.

Line 483-484: The lower… in this region. I don't understand how you conclude this. I don't see this from the analyses. Is this speculation or conclusion?

It has been re-written according to the analyses in section 4 (l. 619-622): "Strong cooling in the Admunsen Sea weakens the transport of relatively less depleted water vapor (compared to the large cooling) inland West Antarctica. It slightly increases the temporal slopes at the WDC and Byrd sites. At the same time, this water vapor contributes to nearby coastal region, decreasing the temporal slopes there (left map of Figure 13)."

Line 498-499: What is meant by the middle of the AMOC peak? I don't understand what is meant here.

Rephrased (l. 644-645): "…because the strong phase period was selected from within the middle of the AMOC peak." See also the Figure S1 and our response to the question for l. 127 in the original manuscript.

Line 506: This comparison to other slopes is not meaningful, as you don't evaluate mean annual temperatures and those studies do.

We use now the annual mean temperatures for the calculation of the temporal slopes (see our response to comment 2). We compare our modeled slopes to the reconstructed ones in section 5. For Antarctica, ECHAM6-wiso simulates too low slopes in Antarctica. For Greenland, it is generally in good agreement from previous reconstructions (Buizert et al., 2014). See our response comment 4.

**References**

Buizert, C., Gkinis, V., Severinghaus, J. P., He, F., Lecavalier, B. S., Kindler, P., Leuenberger, M., Carlson, A. E., Vinther, B., Masson- Delmotte, V., White, J. W. C., Liu, Z., Otto-Bliesner, B., and Brook, E. J.: Greenland temperature response to climate forcing during the last deglaciation, *Science*, 345, 1177–1180, https://doi.org/10.1126/science.1254961, 2014.

Drumond, A., Taboada, E., Nieto, R., Gimeno, L., Vicente-Serrano, S. M., and López-Moreno, J. I.: A Lagrangian analysis of the present-day sources of moisture for major ice-core sites, *Earth Syst. Dynam.*, 7, 549–558, https://doi.org/10.5194/esd-7-549-2016, 2016.

Eyring, V., Bony, S., Meehl, G. A., Senior, C. A., Stevens, B., Stouffer, R. J., and Taylor, K. E.: Overview of the Coupled Model Intercom- parison Project Phase 6 (CMIP6) experimental design and organization, *Geosci. Model Dev.*, 9, 1937–1958, https://doi.org/10.5194/gmd-9-1937-2016, 2016.

Masson-Delmotte, V., Hou, S., Ekaykin, A., Jouzel, J., Aristarain, A., Bernardo, R. T., Bromwhich, D., Cattani, O., Delmotte, M., Falourd, S., Frezzotti, M., Gallée, H., Genoni, L., Isaksson, E., Landais, A., Helsen, M., Hoffmann, G., Lopez, J., Morgan, V., Motoyama, H., Noone, D., Oerter, H., Petit, J. R., Royer, A., Uemura, R., Schmidt, G. A., Schlosser, E., Simões, J. C., Steig, E., Stenni, B., Stievenard, M., van den Broeke, M., van de Wal, R., van de Berg, W.-J., Vimeux, F., and White, J. W. C.: A review of Antarctic surface snow isotopic composition: observations, atmospheric circulation and isotopic modelling, *J. Climate*, 21, 3359–3387, https://doi.org/10.1175/2007JCLI2139.1, 2008.

Sodemann, H., and Stohl, A.: Asymmetries in the moisture origin of Antarctic precipitation. *Geophys. Res. Lett.*, 36(22), L22803, https://doi.org/10.1029/2009GL040242, 2009.

---

## Author Response (AR2)

We acknowledge the 2 reviewers and the editor for their appreciation of our work to revise the manuscript. We thank them again for their reviews and comments that helped to improve our manuscript. We have revised it considering the minor comments detailed below. For the corrections, we provide the line numbers from the revised manuscript (clean version).

**Response to the comment of Editor**

Dear Dr. Cauquoin and co-authors,
Following the referees' comments your revised manuscript needs only some technical/minor revisions. The revised version has very much improved. Please, adjust the manuscript considering these final adjustments suggested by both referees.

We thank the editor for her appreciation of our work on the revised manuscript. We adjusted the manuscript considering the minor comments of the 2 reviewers below.

**Response to the comments of Reviewer 1**

I found the R1 manuscript by Cauquoin et al. largely improved after revision. The authors carefully addressed my (few) and reviewer#2 (more extended) comments. While still long, I believe the manuscript improved especially in readability, e.g. by highlighting the main processes behind each simulation set. I also really enjoyed reading the new discussion part. It is excellent that slopes now are more consistent with observations because of change in the use of annual T instead of P-weighted T.

The aim of the study and its limits are discussed very well, in my opinion. Therefore, my previous positive comment at initial submission stage doesn't change after this revision. I find this manuscript and how the results are presented relevant for CP audience. Therefore, I suggest to accept the paper for publication. I only have a few minor comments about terms/typos, reported hereafter:

We thank the reviewer for his/her appreciation of the revised version of our manuscript.

L367. Please rephrase, very hard to read/understand. Maybe something like "Water vapor d18 in coastal and.... is controlled by nearby local sources"
This sentence has been rephrased (l .367-368): "Water vapor $\delta^{18}O$ in coastal and western low-elevated sites is controlled by nearby local sources, while evaporative moisture source of high-elevation East Antarctic ice cores is typically further north, around 40-45° S (Sodemann and Stohl, 2009)."

L402 I believe it would be more correct to say "a decrease of transport of enriched water vapor" or something similar, rather than speaking of concentration.
Indeed. The sentence has been corrected (l. 404-405): "This can be explained by a decrease of transport of enriched water vapor from the Indian Ocean…"

L473 The word "changed" is repeated.
Corrected (l. 472): "The transport of moisture to Antarctica is generally only slightly changed with variations…"

**Response to the comments of Reviewer 2**

The authors have done a lot of work to improve their paper. The manuscript is in a much better place, with improved analysis and discussion.
I have only some minor technical comments for the authors to address (below.)

We thank the reviewer for his/her appreciation of our efforts to improve the manuscript. His/her comments helped to improve significantly our manuscript.

Line-by-line comments. Important: Line numbers are from the "marked changes" document, not the clean version!

Line 25: change various to variable
Done (l. 25).

Line 462: ice core site location and elevation.
Done (l. 370).

Line 474: Maybe specify here than an increase in the isotope slope means that the imposed changes impact the water isotopes more strongly than the temperatures – correct?
Done (l. 379-381): "…temporal slope values higher between 0 and 90°E longitude compared to the other simulations (meaning that $\delta^{18}O_p$ is more impacted than temperature)."

Line 481-484: I do not understand how you know the d18O of water from the Amundsen sea without having moisture tagging? Please clarify.
We know the $\delta^{18}O$ of water vapor everywhere in the globe. This part has been clarified (l. 388-393): "A larger SST cooling near the Amundsen Sea (i.e., Tierney et al. SST compared to GLOMAP, Figure 5c) impacts the temperature from this region to western Antarctic sites (2 to 4 °C, left map of Figure 5c). On the other hand, the $\delta^{18}O$ of water vapor and precipitation in the Amundsen Sea area is not so impacted by imposed stronger SST cooling (by 2 ‰ at maximum, right map of Figure 5c). The decrease of the transport of this not so depleted water vapor to western Antarctic sites (Figure S7) increases the temporal slopes by ~0.1 % °C$^{-1}$ at WDC and Byrd stations (orange marker in Figure 12)."

Line 485: "making decrease" should be "decreasing"
Corrected (l. 394).

Line 490: What is the "eastern part" of the Southern Ocean? Do you mean Indian ocean sector?
Yes (l. 398-399): "A more extensive sea ice in the Atlantic and Indian sectors of the Southern Ocean changes the transport…"

Line 493: More extensive sea ice where?
It is now specified (l. 402): "The more extensive sea ice in the Indian sector of the Southern Ocean…"

Line 495 "and south of Australia"?
Corrected (l. 404): "…from the Indian Ocean and marine region at south of Australia…"

Line 519: North America instead of Northern USA?
Ok (l. 427).

Line 590: conclusions section: there is some repetition here from section 4 that could be removed I think.
We shortened the first paragraph of the conclusion section.

Line 598-599: "changed" twice in sentence. Remove one.
Corrected (l. 472): "The transport of moisture to Antarctica is generally only slightly changed with variations…"

Line 601: "We found that temporal slopeS…" (make plural)
Corrected (l. 474).

Line 604: like before, what does the south of Australia mean? Do you mean south of Australia, or actual moisture coming off the land area?
Corrected (l. 475-476): "marine region at south of Australia"

Line 671: "if ECHAM6-wiso showS biases…"
Corrected (l. 518).

Line 675: "orographic effects" or "effects of orography"
Effects of orography (l. 522).

Line 679: Yes, I agree. The Antarctic inversion is probably involved.
This is quite a common issue in AGCMs.

Line 697: Maybe an additional conclusion could be that you study shows that the isotopic slopes in Antarctica in model simulations are extremely sensitive to ocean boundary conditions. Therefore, it is dangerous to rely on models to find the slopes one uses in interpreting isotope data, as has been done in the past (for example, Jouzel et al. 1997, 2003, Markle et al. 2022)
Done (l. 539-541): "Finally, by showing the sensitivity of $\delta^{18}O_p$-$T_{2m}$ temporal slopes to sea surface boundary conditions, the potential uncertainties of the latter could have an impact on the reconstruction of the former (Jouzel et al., 1997, 2003; Markle and Steig, 2022)."